# Activation of GCN2/ATF4 signals in amygdalar PKC-δ neurons promotes WAT browning under leucine deprivation

Feixiang Yuan [1], Haizhou Jiang [1], Hanrui Yin [1], Xiaoxue Jiang[1], Fuxin Jiao [1], Shanghai Chen [1], Hao Ying [1], Yan Chen [1], Qiwei Zhai [1] & Feifan Guo [1✉]

The browning of white adipose tissue (WAT) has got much attention for its potential beneficial effects on metabolic disorders, however, the nutritional factors and neuronal signals involved remain largely unknown. We sought to investigate whether WAT browning is stimulated by leucine deprivation, and whether the amino acid sensor, general control nonderepressible 2 (GCN2), in amygdalar protein kinase C-δ (PKC-δ) neurons contributes to this regulation. Our results show that leucine deficiency can induce WAT browning, which is unlikely to be caused by food intake, but is largely blocked by PKC-δ neuronal inhibition and amygdalar GCN2 deletion. Furthermore, GCN2 knockdown in amygdalar PKC-δ neurons blocks WAT browning, which is reversed by over-expression of amino acid responsive gene activating transcription factor 4 (ATF4), and is mediated by the activities of amygdalar PKC-δ neurons and the sympathetic nervous system. Our data demonstrate that GCN2/ATF4 can regulate WAT browning in amygdalar PKC-δ neurons under leucine deprivation.

---

[1] CAS Key Laboratory of Nutrition, Metabolism and Food Safety, Shanghai Institute of Nutrition and Health, University of Chinese Academy of Sciences, Chinese Academy of Sciences, Shanghai, China. ✉email: ffguo@sibs.ac.cn

There are two types of adipose tissue in humans, consisting of white adipose tissue (WAT) and brown adipose tissue (BAT)[1]. WAT stores vast amounts of chemical energy as triglycerides and BAT stimulates thermogenesis via increased uncoupling protein-1 (UCP1) expression[1]. Under certain conditions, WAT can change to BAT-like adipocytes via a process called as WAT browning, and such cells are defined as beige fat cells[2]. In addition to being stimulated by cold exposure[3], WAT browning may also be inhibited by continuous fasting[4], or stimulated by intermittent fasting or a low-protein/high carbohydrate diet[5,6], suggesting that nutrients may also be an important external stimulus for this regulation. Amino acids are building blocks of protein. However, increasing evidence suggests that they are also critical metabolic signals for many important functions[7,8]. For example, elevated levels of branched-chain amino acids (BCAAs), including leucine, valine, and isoleucine, are closely related to lipid metabolism, glucose utilization, and energy expenditure[9,10]. It has also been shown that leucine deprivation decreases fat mass, which is associated with increased WAT lipolysis and BAT thermogenesis[11]. However, it remains unknown whether WAT browning is induced on this condition.

Activation of the sympathetic nervous system (SNS) is one of the major effectors mediating the effects of various external stimuli, such as cold exposure or exercise, on WAT browning[12,13]. The hypothalamus and certain distinct hypothalamic neurons including agouti-related protein (AgRP) and proopiomelanocortin (POMC) are involved in this regulation[14,15]. The neuronal control of WAT browning by other brain areas, however, remains largely unknown. The amygdala is an almond-shaped group of nuclei at the heart of the telencephalon that consists of three main parts, the central amygdala (CeA), the basolateral amygdala (BLA), and the medial amygdala (MeA)[16]. Though the amygdala is well-known to be associated with many cognitive functions, such as memory and social behavior[17], recent studies have shown that the amygdala is also involved in the regulation of some metabolic processes[18–20]. For example, protein kinase C-δ

(PKC-δ) neurons in the CeA are reported to mediate anorexigenic signals[21]. These results suggest that amygdalar PKC-δ neurons may have other metabolic functions that need to be further explored.

General control nonderepressible 2 (GCN2) is an ancient protein kinase that senses intracellular amino acid deficiencies, which then couples the accumulation of uncharged transfer RNAs (tRNAs) to the phosphorylation of eukaryotic initiation factor 2α (eIF2α). This thereby increases the translation of mRNAs for several effectors that have many functions, such as regulating amino acid biosynthesis and transport, as adaptive responses[22]. One such example is the amino acid response gene-activating transcription factor 4 (ATF4)[22]. It has been shown that GCN2 plays an important role in metabolic responses during protein restriction and methionine restriction[23,24] and regulates insulin sensitivity under states of leucine deprivation[25]. Therefore, it is likely that GCN2/ATF4 signaling is involved in leucine deprivation-induced other metabolic effects, such as WAT browning.

The aim of this study was to investigate whether WAT browning is stimulated by leucine deprivation. Our results suggest that leucine deprivation can induce WAT browning, a process that is regulated by GCN2/ATF4 signaling in amygdalar PKC-δ neurons.

## Results

**Leucine deprivation induces WAT browning.** To investigate whether leucine deprivation promotes WAT browning, we fed wild-type (WT) mice a control or leucine-deficient diet for 3 days. Similar to mice subjected to leucine deprivation for 7 days[11], mice subjected to 3 days of leucine deprivation decreased food intake and had a significantly reduced body fat mass, including subcutaneous WAT (sWAT) and epididymal WAT (eWAT), compared with mice fed a control diet (Fig. 1a–c and Supplementary Fig. 1a). Furthermore, a distinct histological morphology with the presence of multilocular small lipid droplets [as demonstrated by

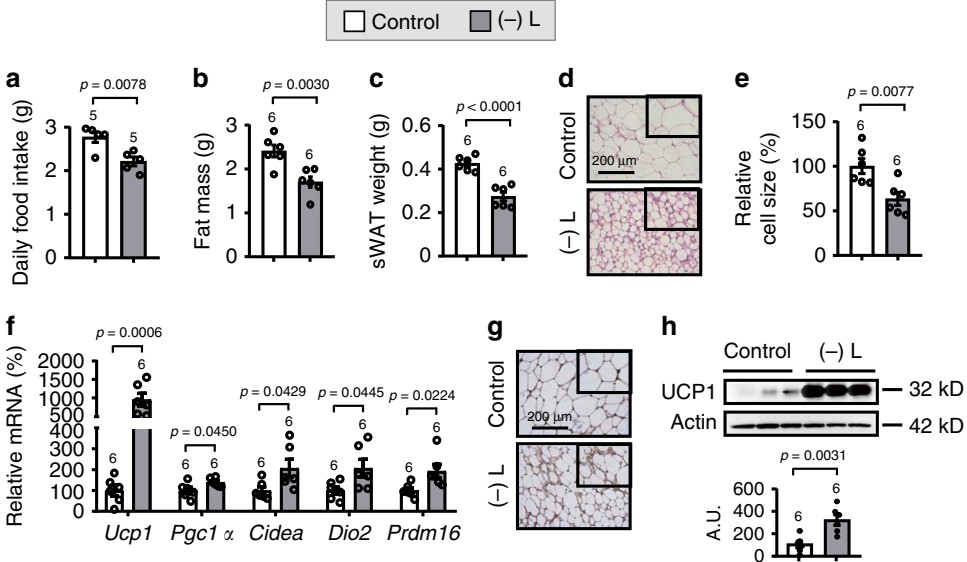

**Fig. 1 Leucine deprivation induces WAT browning. a** Daily food intake. **b** Fat mass by NMR. **c** Subcutaneous WAT (sWAT) weight. **d** Representative images of hematoxylin and eosin (H&E) staining of sWAT. **e** sWAT cell size quantified by Image J analysis of H&E images. **f** Gene expression of *Ucp1*, *Pgc1a*, *Cidea*, *Dio2*, and *Prdm16* in sWAT by RT-PCR. **g** Representative images of immunohistochemistry (IHC) staining of UCP1 in sWAT. **h** UCP1 protein in sWAT by western blotting (top) and quantified by densitometric analysis (bottom); A.U.: arbitrary units. Studies were conducted using 14- to 15-week-old male wild-type mice fed a control (Control) or leucine-deficient [(-) L] diet for 3 days. Data are expressed as the mean ± SEM (*n* represents number of samples and are indicated above the bar graph), with individual data points. Data were analyzed by two-tailed unpaired Student's *t* test. Source data are provided as a Source data file.

hematoxylin and eosin (H&E) staining and evaluated by cell size], which is the characteristic of browning fat[1], was observed in the sWAT of leucine-deprived mice (Fig. 1d, e). To further confirm the phenomenon, we examined the expression of genes related to WAT browning, including *uncoupling protein-1 (Ucp1), peroxisome proliferator-activated receptor gamma co-activator 1α (Pgc1α), cell death-inducing DFFA-like effector a (Cidea), deiodinase 2 (Dio2)*, and *PR domain containing 16 (Prdm16)*[14,15,26], in the sWAT of these mice in response to different diets. The expression of WAT browning markers measured by RT-PCR, as well as UCP1 protein, as assessed by immunohistochemistry (IHC) and western blotting analysis, were all increased in the sWAT of leucine-deprived mice (Fig. 1f–h). Similar results were obtained in eWAT of these mice (Supplementary Fig. 1b–f). Moreover, the effect of leucine deprivation on WAT browning started after the first day of leucine deprivation and lasted for the whole period for the experiment (Supplementary Fig. 2).

**Inhibition of amygdalar PKC-δ neurons blocks WAT browning.** To investigate the possible contribution of the amygdala in leucine deprivation-induced WAT browning, we conducted immunofluorescence (IF) staining to examine the changes of c-Fos, a signal reflecting neuronal activity[21], in the amygdala of WT mice fed a control or leucine-deficient diet. We found that leucine deprivation increased c-Fos staining in several areas of amygdala, including the CeA which is involved in metabolic responses to the amino acid imbalanced diet[27] and BLA (Supplementary Fig. 3a). We next assessed whether PKC-δ neurons in the CeA[21,28] were involved in leucine deprivation-induced WAT browning by examining c-Fos staining in these neurons in PKC-δ-Cre-Ai9 mice. As predicted, IF staining of tdTomato (reflecting PKC-δ neurons) and c-Fos revealed that c-Fos levels were increased in the PKC-δ neurons of leucine-deprived mice (Fig. 2a), suggesting that the activity of amygdalar PKC-δ neurons was increased in response to leucine deprivation. We then tested the effect of chemogenetically inhibition of amygdalar PKC-δ neuronal activity on leucine deprivation-induced WAT browning, using an inhibitory hM4Di designer receptors exclusively activated by designer drugs (DREADDs), which are activated by the inert ligand clozapine N-oxide (CNO)[21]. To this end, a Cre-dependent adeno-associated virus (AAV) encoding hM4Di (AAV-DIO-hM4Di-mCherry) or mCherry (AAV-DIO-mCherry) was injected into the CeA of PKC-δ-Cre mice, and all of these mice were then intraperitoneally (i.p.) injected with CNO 4 weeks after AAV delivery. The inhibited PKC-δ neuronal activity was then confirmed by the reduced IF staining of c-Fos in PKC-δ neurons (reflected by mCherry) in mice injected with hM4Di (Supplementary Fig. 3b). Inhibiting the neuronal activity of PKC-δ neurons partly influenced the food intake reduction and largely blocked leucine deprivation-induced WAT browning, as demonstrated by the corresponding changes in fat mass, sWAT weight, H&E staining, cell size, as well as the expression of markers for browning in sWAT (Fig. 2b–i). Similar changes in the weight, H&E staining, and cell size of eWAT were observed in these mice (Supplementary Fig. 3c–e).

**Leucine deprivation activates GCN2 to induce WAT browning.** GCN2 is an amino acid sensor expressed in many areas of brain, including the amygdala (Allen Brain Atlas). In agreement with the previous reports of 7 days' leucine deprivation[25], leucine deprivation for 3 days also decreased serum and amygdalar leucine levels compared with mice under a control diet (Supplementary Fig. 4), implying the possibility of the activation and involvement of GCN2 in this regulation. To assess whether GCN2 was involved in the regulation of WAT browning in response to

leucine deprivation, we tested whether WAT browning was blocked in global GCN2 knockout mice. We found that leucine deprivation-induced WAT browning was blocked in global GCN2 knockout mice (Supplementary Fig. 5). The observation that phosphorylation of GCN2 (p-GCN2) and its downstream target eIF2α (p-eIF2α) were elevated in the amygdala of leucine-deprived mice pointed to a possible role of amygdalar GCN2 in the regulation of WAT browning (Fig. 3a).

If GCN2 activation in the amygdala was responsible for WAT browning under leucine deprivation, knockdown of amygdalar GCN2 would be expected to exert a blocking effect. To test this hypothesis, we stereotaxically injected AAVs expressing Cre-GFP or GFP into the amygdala of GCN2 loxp/loxp (GCN2$^{+/+}$) mice (Supplementary Fig. 6a), either to delete GCN2 exclusively in the amygdala (GCN2 KO) or act as control, respectively, and then maintained these mice on a control or leucine-deficient diet for 3 days. RT-PCR and western blot analyses revealed that GCN2 levels were decreased by more than 50% in the amygdala, but not in other brain areas [such as the arcuate nucleus (ARC) of the hypothalamus] in GCN2 KO mice (Supplementary Fig. 6b and c). Furthermore, GCN2 was detected in GFP-infection neurons in control mice, but absent in GCN2 KO mice (Supplementary Fig. 6d). The knockdown efficiency of GCN2 was further confirmed by the inhibited expression of its downstream effector ATF4 and a few ATF4-dependent genes[29] in the amygdala of GCN2 KO mice (Supplementary Fig. 6e).

We then examined the effect of amygdalar GCN2 deletion on WAT browning under leucine deprivation. The reduced food intake under leucine deprivation was partly reversed in GCN2 KO mice (Fig. 3b). Moreover, deletion of GCN2 in the amygdala blocked leucine deprivation-induced WAT browning, which was demonstrated by the corresponding changes in fat mass, sWAT weight, H&E staining, cell size, as well as the expression of markers for WAT browning in sWAT (Fig. 3c–i). Similar results were obtained in the eWAT of these mice (Supplementary Fig. 6f–k). Leucine deprivation-regulated lipolysis related gene and proteins[30] were also blocked in the sWAT of GCN2 KO mice (Supplementary Fig. 7). In contrast, no differences were observed in terms of the thermogenesis-related parameters examined in BAT between two strains of mice under leucine deprivation (Supplementary Fig. 8). Moreover, deletion of GCN2 in the amygdala partly prevented the reduction in body weight, the increase in oxygen consumption and the energy expenditure by leucine deprivation, though it had no significant effect on the respiratory exchange ratio (RER; $V_{CO_2}/V_{O_2}$), physical activity, or rectal temperature under leucine deprivation (Supplementary Fig. 9).

**PKC-δ neuron activation reverses effects of GCN2 knockdown.** We subsequently investigated whether GCN2 in amygdalar PKC-δ neurons was responsible for leucine deprivation-induced WAT browning. The increased IF staining of p-GCN2 and p-eIF2α in the CeA of WT mice (Supplementary Fig. 10a) and the increased staining of p-GCN2 in the PKC-δ neurons (reflected by tdTomato) of PKC-δ-Cre-Ai9 mice (Fig. 4a) under leucine deprivation pointed to a possible role of GCN2 in these neurons. We next investigated whether knockdown of GCN2 in amygdalar PKC-δ neurons would block leucine deprivation-induced WAT browning. To this end, Cre-dependent AAVs encoding a short hairpin RNA (shRNA) directed against GCN2 (AAV-Flex-shGCN2-GFP) or GFP (AAV-Flex–GFP) were injected into the CeA of PKC-δ-Cre mice (Supplementary Fig. 10b). IF staining of GFP (reflecting PKC-δ neurons) and GCN2 revealed that GCN2 was colocalized with PKC-δ neurons in control mice, but was significantly reduced in PKC-δ-shGCN2 mice (Supplementary Fig. 10c).

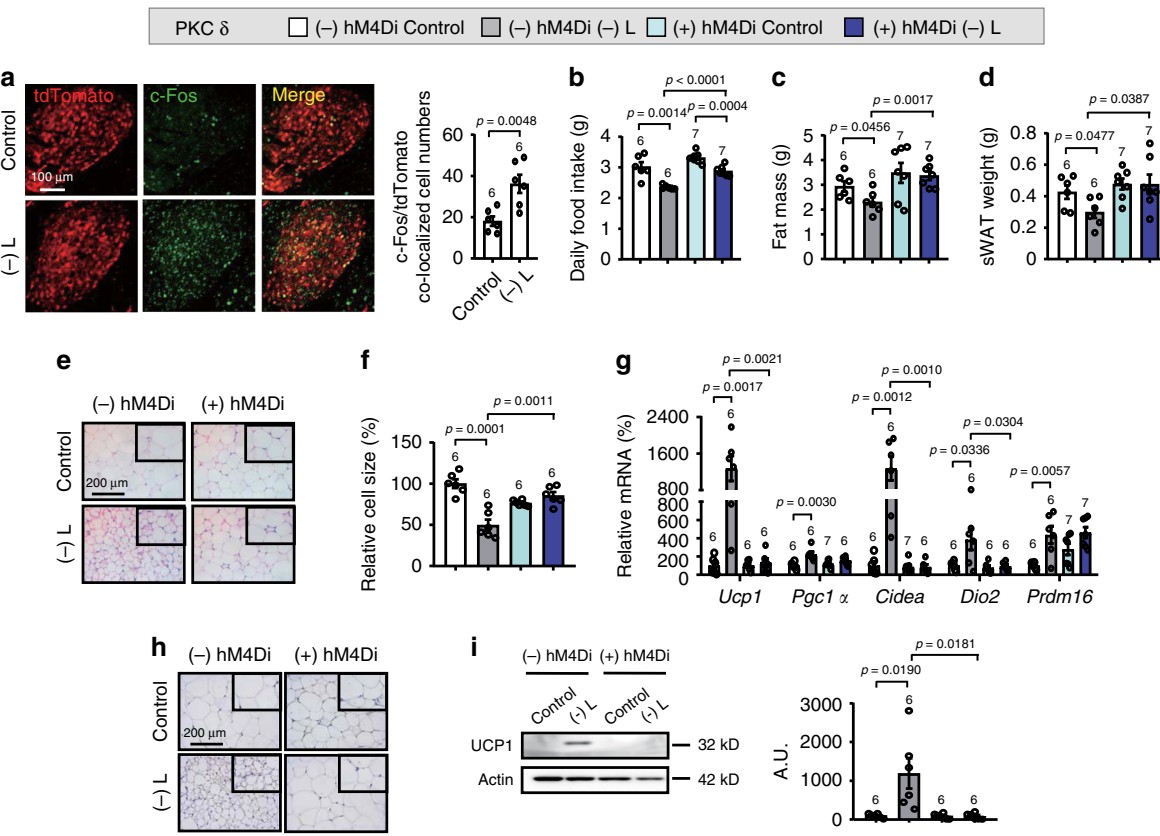

**Fig. 2 Silencing PKC-δ neuronal activity blocks WAT browning under leucine deprivation. a** Immunofluorescence (IF) staining for tdTomato (red), c-Fos (green) and merge (yellow) in central amygdala (CeA) sections (left), and quantification of c-Fos and tdTomato colocalized cell numbers (right). **b** Daily food intake. **c** Fat mass by NMR. **d** Subcutaneous WAT (sWAT) weight. **e** Representative images of hematoxylin and eosin (H&E) staining of sWAT. **f** sWAT cell size quantified by Image J analysis of H&E images. **g** Gene expression of *Ucp1*, *Pgc1a*, *Cidea*, *Dio2*, and *Prdm16* in sWAT by RT-PCR. **h** Representative images of immunohistochemistry (IHC) of UCP1 in sWAT. **i** UCP1 protein in sWAT by western blotting (left) and quantified by densitometric analysis (right); A.U.: arbitrary units. Studies for **a** were conducted using 12- to 14-week-old male PKC-δ-Cre-Ai9 mice fed a control (Control) or leucine-deficient [(-) L] diet for 3 days; studies for **b**–**i** were conducted using 22- to 24-week-old male PKC-δ-Cre mice receiving AAVs expressing mCherry (PKC δ − hM4Di) or hM4Di (PKC δ + hM4Di), all received CNO injections every 12 h for 3 days, simultaneously fed a Control or (-) L diet for 3 days. Data are expressed as the mean ± SEM (*n* represents number of samples and are indicated above the bar graph), with individual data points. Data were analyzed by two-tailed unpaired Student's *t* test for **a**, or by one-way ANOVA followed by the SNK (Student–Newman–Keuls) test for **b**–**i**. Source data are provided as a Source data file.

However, the expression of GCN2 in BLA remained unchanged between both strains of mice (Supplementary Fig. 10c). Under leucine deprivation, the decreased food intake was reversed to some extent in PKC-δ-shGCN2 mice (Fig. 4b). More importantly, knockdown of GCN2 in PKC-δ neurons blocked leucine deprivation-induced WAT browning, as demonstrated by the corresponding changes in fat mass, sWAT weight, H&E staining, cell size, as well as the expression of markers for WAT browning in sWAT (Fig. 4c–i). Similar results were obtained in the eWAT of these mice (Supplementary Fig. 10d–h).

Because the increased c-Fos expression (reflecting increased neuronal activity) in the amygdala under leucine deprivation was largely blunted by deletion of GCN2 in the amygdala (Supplementary Fig. 11), we speculated that the blocking effect of GCN2 might be due to the inhibited neuronal activity. We then tested whether activation of PKC-δ neurons by an excitatory DREADD receptor hM3Dq activated by CNO[31] would reverse the blocking effect of GCN2 knockdown on leucine deprivation-induced WAT browning. For this purpose, Cre-dependent-AAVs encoding hM3Dq (AAV-DIO-hM3Dq-mCherry) or mCherry (AAV-DIO-mCherry) were injected into the CeA of PKC-δ-Cre mice in which GCN2 was knocked down in these neurons. All of these mice were then i.p. injected with CNO 4 weeks after AAV delivery. As expected, activation of PKC-δ neurons reversed the blocking effect of GCN2 knockdown in amygdalar PKC-δ neurons on leucine deprivation-induced WAT browning, as reflected in all the related parameters assessed, as mentioned above (Supplementary Fig. 12).

**ATF4 in amygdalar PKC-δ neurons regulates WAT browning.** Previous studies have shown that activation of the GCN2/eIF2α pathway increases the translation of several mRNAs, including ATF4, that have many different functions under leucine deprivation[22,32,33]. We speculated that ATF4 might function as a downstream effector of GCN2 in the regulation of WAT browning under such conditions. Consistent with this possibility, IF staining revealed that ATF4 was increased in amygdalar PKC-δ neurons, as assessed in PKC-δ-Cre-Ai9 mice under leucine deprivation (Supplementary Fig. 13a). RT-PCR analysis showed that the expression of *Atf4* and its downstream target gene *tribbles homolog 3* (*Trb3*)[34] were increased in the amygdala of leucine-deprived mice (Fig. 5a), and that this increase was blocked in mice in whom GCN2 was knocked down in PKC-δ neurons (Supplementary Fig. 13b, c).

Based on the above results, we tested whether inhibition of ATF4 in amygdalar PKC-δ neurons would block leucine

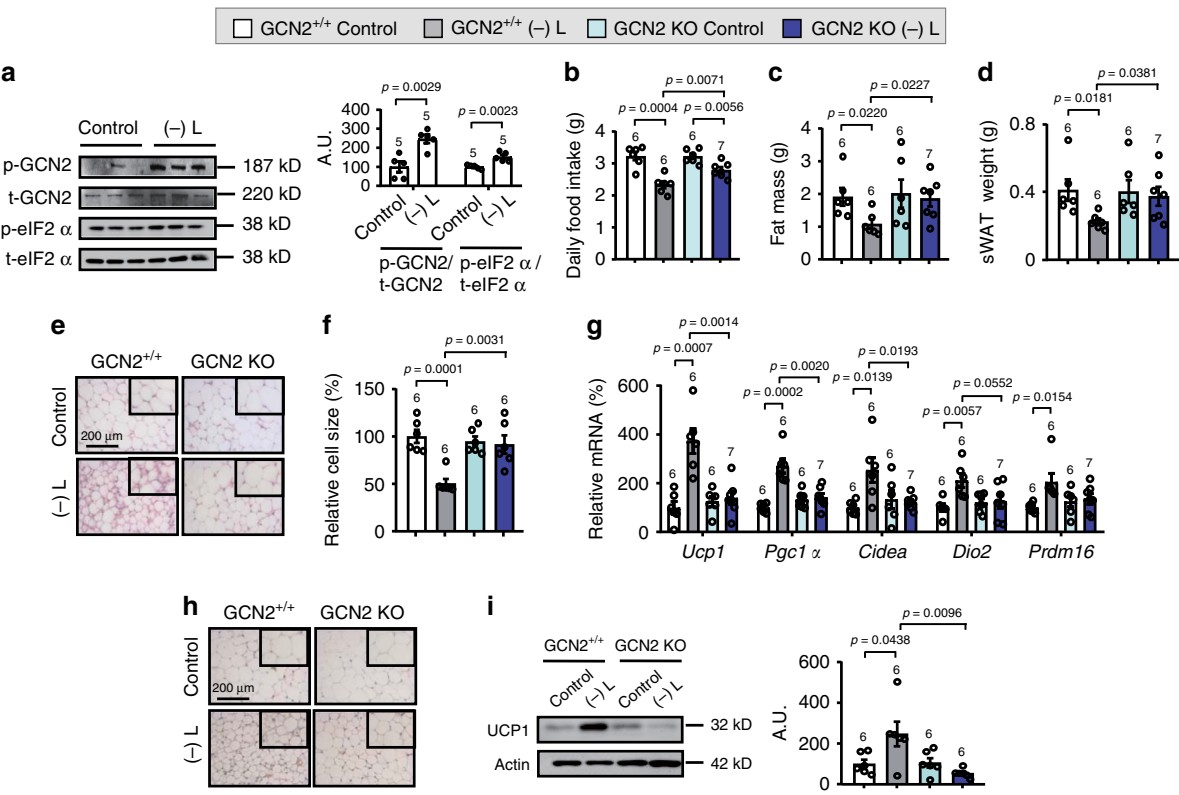

**Fig. 3 GCN2 deletion in amygdala markedly blunts leucine deprivation-induced WAT browning. a** P-GCN2, t-GCN2, p-eIF2α, and t-eIF2α proteins in amygdala by western blotting (left) and quantified by densitometric analysis (right); A.U.: arbitrary units. **b** Daily food intake. **c** Fat mass by NMR. **d** Subcutaneous WAT (sWAT) weight. **e** Representative images of hematoxylin and eosin (H&E) staining of sWAT. **f** sWAT cell size quantified by Image J analysis of H&E images. **g** Gene expression of *Ucp1, Pgc1a, Cidea, Dio2*, and *Prdm16* in sWAT by RT-PCR. **h** Representative images of immunohistochemistry (IHC) staining of UCP1 in sWAT. **i** UCP1 protein in sWAT by western blotting (left) and quantified by densitometric analysis (right). All studies were conducted using 20- to 22-week-old male control mice (GCN2$^{+/+}$) or mice with GCN2 deletion in amygdala (GCN2 KO) fed a control (Control) or leucine-deficient [(-) L] diet for 3 days. Data are expressed as the mean ± SEM (*n* represents number of samples and are indicated above the bar graph), with individual data points. Data were analyzed by two-tailed unpaired Student's *t* test for (**a**), or by one-way ANOVA followed by the SNK (Student–Newman–Keuls) test for (**b-i**). Source data are provided as a Source data file.

deprivation-induced WAT browning. To this end, we injected Cre-dependent AAVs encoding a dominant-negative form of ATF4 (DN ATF4)[34,35], which is AAV-DIO-DN-ATF4-mCherry, or mCherry (AAV-DIO-mCherry) into the CeA of PKC-δ-Cre mice (Supplementary Fig. 13d), and fed these mice either a control or leucine-deficient diet. AAV-mediated DN ATF4 was confirmed by the increased ATF4 and decreased TRB3 expression in amygdala, but not in other brain areas (such as the hypothalamus) (Supplementary Fig. 13e–g). During leucine deprivation, mice received DN-ATF4 AAVs ate more food than control mice, though very limited (Fig. 5b). And as expected, inhibition of ATF4 in PKC-δ neurons blocked leucine deprivation-induced browning in sWAT, as demonstrated by the corresponding changes in fat mass, sWAT weight, H&E staining, cell size, as well as the expression of markers for WAT browning (Fig. 5c–i). Similar results were obtained in eWAT of these mice (Supplementary Fig. 13h–l).

We then tested whether activation of ATF4 in PKC-δ neurons would mimic leucine deficiency-induced WAT browning. To this end, we injected Cre-dependent AAVs expressing ATF4 (AAV-DIO-ATF4-mCherry) or mCherry (AAV-DIO-mCherry) into the CeA of PKC-δ-Cre mice (Supplementary Fig. 14a). Overexpression of ATF4 in the amygdala, but not in other brain areas (such as the hypothalamus), was confirmed by RT-PCR and western blotting analysis (Supplementary Fig. 14b, c). Increased ATF4 in amygdalar PKC-δ neurons was then demonstrated by IF staining (Supplementary Fig. 14d). Metabolic parameters were then

analyzed in these mice. The fat mass, as well as the tissue weight and cell size of sWAT and eWAT, were decreased in PKC-δ-ATF4 mice compared with control mice (Supplementary Fig. 14e–h). Consistently, the browning markers were markedly increased in sWAT and eWAT of these mice (Supplementary Fig. 14i–m).

We then assessed whether activation of ATF4 in amygdalar PKC-δ neurons could reverse the blocking effect of GCN2 knockdown on leucine deprivation-induced WAT browning. To test this possibility, we injected Cre-dependent AAVs expressing shGCN2 (AAV-Flex-shGCN2-GFP) and ATF4 (AAV-DIO-ATF4-mCherry), as well as control AAVs, as indicated, into the CeA of PKC-δ-Cre mice. We then fed these mice either a control or leucine-deficient diet. We found that overexpression of ATF4 in amygdalar PKC-δ neurons reversed the blocking effect of GCN2 knockdown on leucine deprivation-induced WAT browning, as demonstrated by the corresponding changes in fat mass, adipose tissue weight, H&E staining, cell size, as well as the expression of markers for WAT browning both in sWAT and eWAT (Supplementary Fig. 15).

**SNS activities regulate WAT browning under leucine deficiency.** Next, we investigated how the signals in the brain were delivered to sWAT. Previous studies have shown that activation of SNS induces WAT browning[3]. Here, we found that leucine deprivation activated SNS in sWAT, as shown by the markers

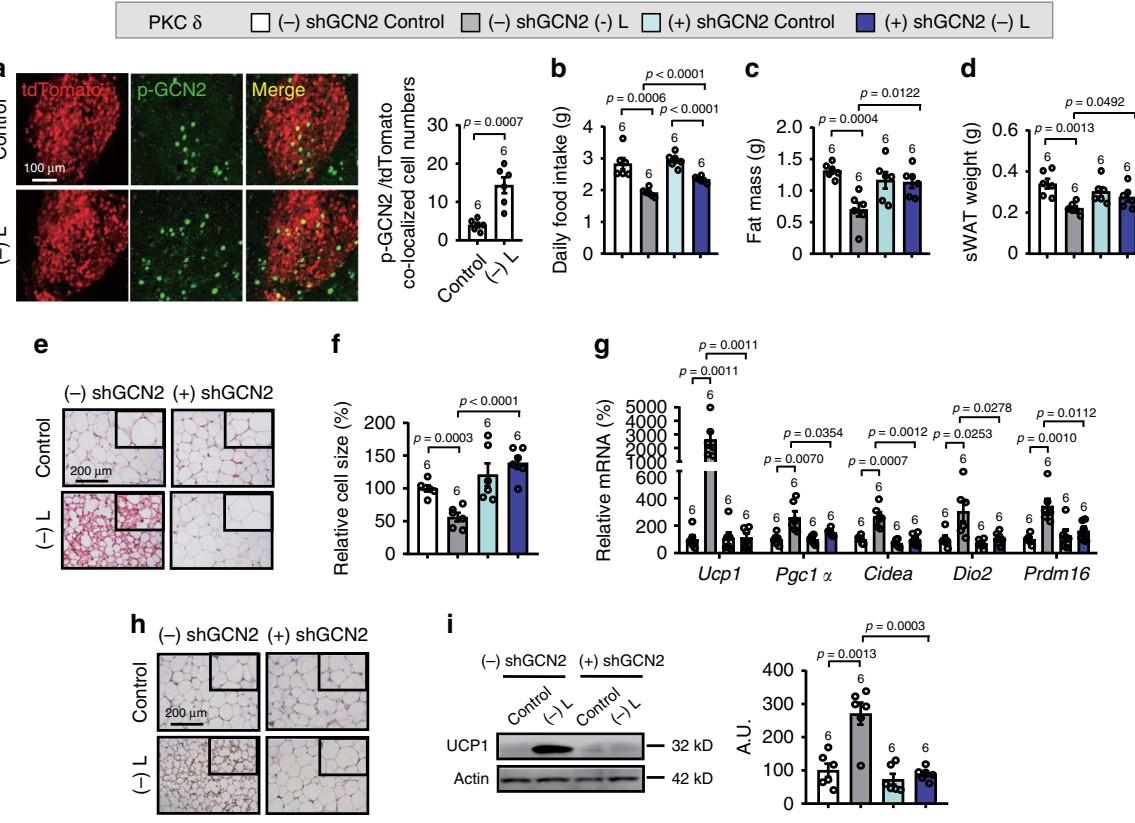

**Fig. 4 Knockdown of GCN2 in amygdalar PKC-δ neurons blocks leucine deprivation-induced WAT browning. a** Immunofluorescence (IF) staining for tdTomato (red), p-GCN2 (green) and merge (yellow) in central amygdala (CeA) sections (left), and quantification of p-GCN2 and tdTomato colocalized cell numbers (right). **b** Daily food intake. **c** Fat mass by NMR. **d** Subcutaneous WAT (sWAT) weight. **e** Representative images of hematoxylin and eosin (H&E) staining of sWAT. **f** sWAT cell size quantified by Image J analysis of H&E images. **g** Gene expression of *Ucp1, Pgc1a, Cidea, Dio2,* and *Prdm16* in sWAT by RT-PCR. **h** Representative images of immunohistochemistry (IHC) of UCP1 in sWAT. **i** UCP1 protein in sWAT by western blotting (left) and quantified by densitometric analysis (right); A.U.: arbitrary units. Studies for **a** were conducted using 12- to 14-week-old male PKC-δ-Cre-Ai9 mice fed a control (Control) or leucine-deficient [(-) L] diet for 3 days; studies for **b–i** were conducted using 13- to 16-week-old male PKC-δ-Cre mice receiving AAVs expressing GFP (PKC δ − shGCN2) or shGCN2 (PKC δ + shGCN2) fed a Control or (-) L diet for 3 days. Data are expressed as the mean ± SEM (*n* represents number of samples and are indicated above the bar graph), with individual data points. Data were analyzed by two-tailed unpaired Student's *t* test for **a**, or one-way ANOVA followed by the SNK (Student–Newman–Keuls) test for **b–i**. Source data are provided as a Source data file.

reflecting activated SNS including the increased levels of nor-epinephrine (NE), tyrosine hydroxylase (TH) protein, and β-adrenergic receptor 3 (*Adrb3*) mRNA[26,36] in sWAT, and these changes were blunted by knockdown of GCN2 or ATF4 in amygdalar PKC-δ neurons (Fig. 6a–c and Supplementary Fig. 16a–i).

To assess the essential role of SNS in the browning process, we applied the pharmacotoxic approach of using 6-hydroxydopamine (6-OHDA) to locally ablate the sympathetic fibers[3] in the sWAT of leucine-deprived mice. Sympathetic denervation (as reflected by the reduced levels of NE, TH protein, and *Adrb3* mRNA in sWAT) attenuated the effects of leucine deprivation on fat mass, sWAT weight, and sWAT browning signals, but had no effect in eWAT (Fig. 6d–j and Supplementary Fig. 16j–l).

To further confirm the role of SNS in amygdalar PKC-δ neurons regulation of leucine deprivation-induced WAT browning, we examined the levels of NE, TH expression and *Adrb3* mRNA in the sWAT of mice in which PKC-δ neuronal activity was inhibited by hM4Di. Notably, leucine deprivation-activated SNS in sWAT was blocked in mice with inhibition of PKC-δ neuronal activity (Fig. 7a–c). If the decreased SNS activity was responsible for PKC-δ neuronal regulation of WAT browning, activation of SNS activity would be expected to reverse this

inhibitory effect. To test this possibility, we i.p. injected the β3-adrenergic agonist CL316, 243[14] or a control vehicle into mice with inhibition of PKC-δ neuronal activity under leucine deprivation. As expected, activation of SNS reversed the blocking effect of PKC-δ neurons' inhibition on leucine deprivation-induced WAT browning, as demonstrated by the measurement of main parameters mentioned above (Fig. 7d–j and Supplementary Fig. 17).

## Discussion

There has been a recent rise in interest in studying the browning of fat, particularly in light of its beneficial effects on metabolic disorders[37–39]. WAT browning is stimulated by cold exposure, exercises or intermittent fasting[6,39]. However, very little is known regarding its nutritional control. In this study, we identify a nutrient, leuicne, which has a significant impact on WAT browning: Dietary leucine deficiency is sufficient to induce WAT browning. These studies demonstrate the important role of nutrition in the regulation of WAT browning and suggest that manipulating dietary nutrients might be a possible strategy to induce WAT browning.

Studies in the past few years have greatly expanded our knowledge of the brain's regulation of WAT browning, especially in the hypothalamus[14,15,26,40]. For example, O-GlcNAc signaling

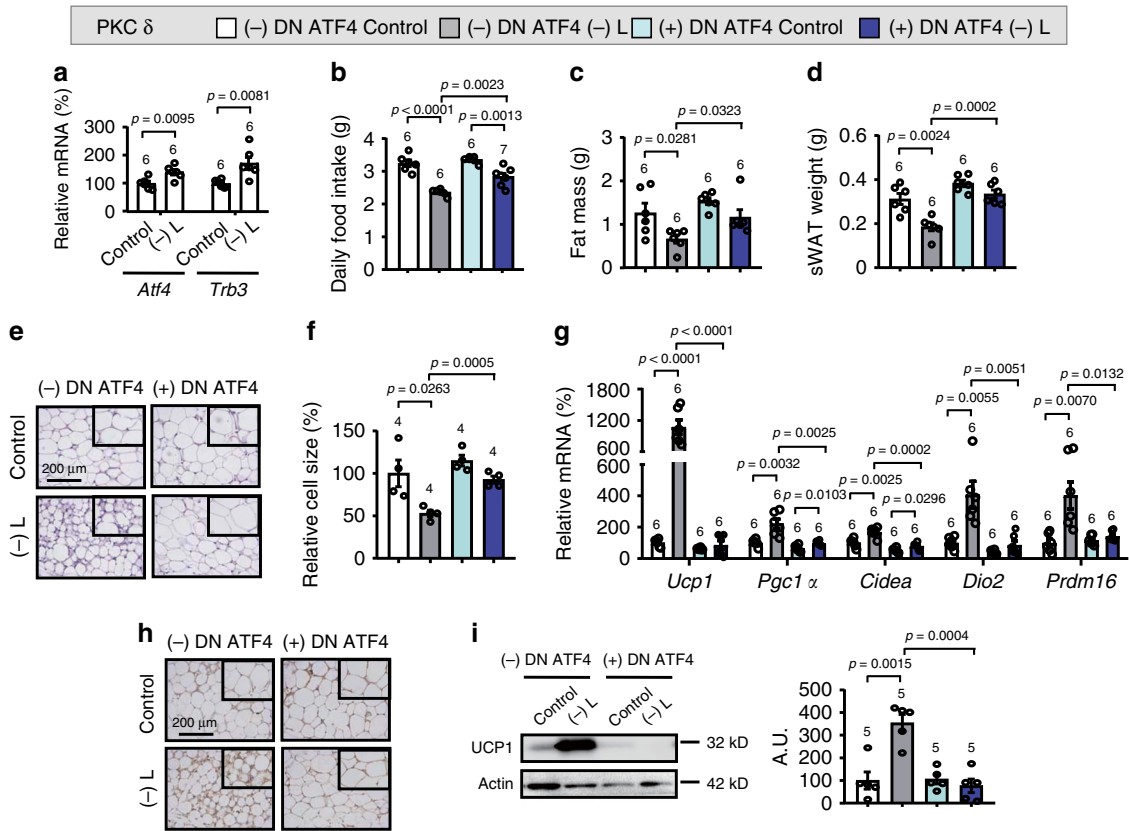

**Fig. 5 Specially inhibition of ATF4 in amygdalar PKC-δ neurons blocks leucine deprivation-induced WAT browning. a** Gene expression of *Atf4* and *Trb3* in amygdala by RT-PCR. **b** Daily food intake. **c** Fat mass by NMR. **d** Subcutaneous WAT (sWAT) weight. **e** Representative images of hematoxylin and eosin (H&E) staining of sWAT. **f** sWAT cell size quantified by Image J analysis of H&E images. **g** Gene expression of *Ucp1, Pgc1a, Cidea, Dio2*, and *Prdm16* in sWAT by RT-PCR. **h** Representative images of immunohistochemistry (IHC) of UCP1 in sWAT. **i** UCP1 protein in sWAT by western blotting (left) and quantified by densitometric analysis (right); A.U.: arbitrary units. Studies for **a** were conducted using 14- to 15-week-old male wild-type mice fed a control (Control) or leucine-deficient [(-) L] diet for 3 days; studies for (**b–i**) were conducted using 13- to 16-week-old male PKC-δ-Cre mice receiving AAVs expressing mCherry (PKC δ − DN ATF4) or DN ATF4 (PKC δ + DN ATF4) fed a Control or (-) L diet for 3 days. Data are expressed as the mean ± SEM (*n* represents number of samples and are indicated above the bar graph), with individual data points. Data were analyzed by two-tailed unpaired Student's *t* test for (**a**), or one-way ANOVA followed by the SNK (Student–Newman–Keuls) test for (**b–i**). Source data are provided as a Source data file.

in AgRP neurons suppresses browning of white fat, while insulin and leptin act on POMC neurons to promote browning[14,15]. However, knowledge of which other brain regions are involved, beyond the hypothalamus, remains very limited. Amygdalar PKC-δ neurons are known to mediate the influence of diverse anorexigenic signals[21]. Besides feeding, PKC-δ neurons also affect emotional states, including fear and anxiety[28,41]. However, whether amygdalar PKC-δ neurons regulate WAT browning has not yet been reported. As shown previously[27], we also found that c-Fos was activated in the amygdala in response to deficiency of essential amino acids, indicating the involvement of amygdala in this regulation. Furthermore, leucine deprivation stimulated the activity of amygdalar PKC-δ neurons, while inhibiting them blocked WAT browning under leucine deprivation. These results point to an important role of amygdalar PKC-δ neurons in the regulation of WAT browning under leucine deprivation. Our finding is the first to document the function of these neurons in the regulation of lipid metabolism in adipose tissue, therefore expanding our understanding of the important functions controlled by PKC-δ neurons. However, leucine deprivation also increases c-Fos expression in BLA. As such, the possible involvement of BLA in the regulation of WAT browning under leucine deprivation remains to be further studied in the future.

It is well-known that GCN2 is an amino acid sensor[22] and that ATF4 plays an important role in sensing amino acid deficiency[33] while taking part in many adaptive responses under various conditions like high-fat diet feeding[42], ER stress, and cancer[43]. This study showed that GCN2 and ATF4 in amygdalar PKC-δ neurons were activated, and that knockdown of GCN2 or inhibition of ATF4 in these neurons blocked leucine deprivation-induced WAT browning. Many metabolic functions are controlled by altered neuronal activity[20,26]. Consistently, we found that the effect of GCN2 in amygdalar PKC-δ neurons under leucine deprivation was mediated by the stimulation of these neurons' activity. These results suggest that GCN2 and ATF4 can regulate WAT browning under leucine deprivation, by altering the neuronal activity of amygdalar PKC-δ neurons.

It has been previously reported that changes in neuronal activity may affect the activity of SNS[12,26], which has been shown to play an important role in WAT browning under different circumstances[3,14,15]. A previous study has shown that leucine deficiency activates SNS as confirmed by increased serum NE levels[11]. In this study, we found that the activated SNS fibers could reach sWAT. The importance of the activated SNS in leucine deprivation-regulated WAT browning was confirmed by the effect of pharmacologic ablation by 6-OHDA-induced

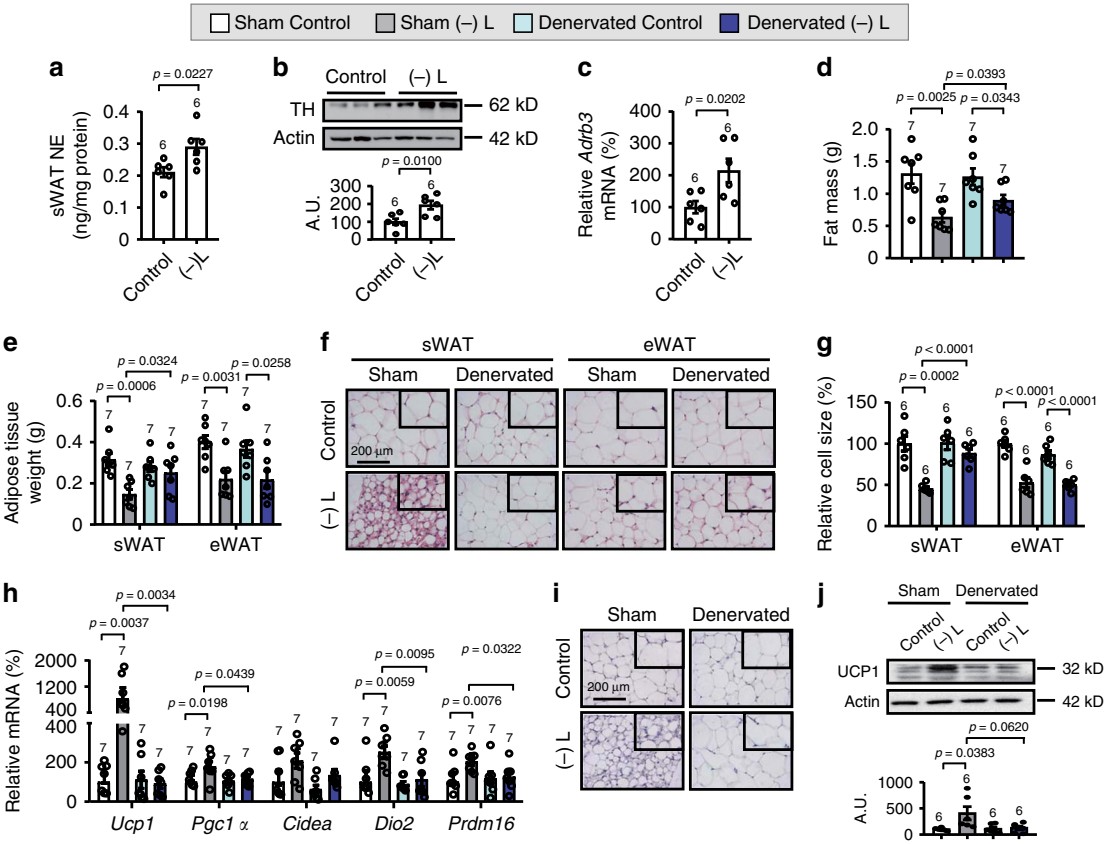

**Fig. 6 Sympathetic denervation in sWAT blocks leucine deprivation-induced WAT browning. a** Norepinephrine (NE) levels in subcutaneous WAT (sWAT) measured by ELISA kit. **b** Tyrosine hydroxylase (TH) proteins in sWAT by western blotting (top) and quantified by densitometric analysis (bottom); A.U.: arbitrary units. **c** Gene expression of *Adrb3* in sWAT by RT-PCR. **d** Fat mass by NMR. **e** Adipose tissue weight. **f** Representative images of hematoxylin and eosin (H&E) staining of sWAT and epididymal WAT (eWAT). **g** Cell size quantified by Image J analysis of H&E images. **h** Gene expression of *Ucp1, Pgc1a, Cidea, Dio2*, and *Prdm16* in sWAT by RT-PCR. **i** Representative images of immunohistochemistry (IHC) of UCP1 in sWAT. **j** UCP1 protein in sWAT by western blotting (top) and quantified by densitometric analysis (bottom). Studies for **a–c** were conducted using 14- to 15-week-old male wild-type (WT) mice fed a control (Control) or leucine-deficient [(-) L] diet for 3 days; studies for **d–j** were conducted using 12-week-old male WT mice with sham operated (Sham) or denervated (Denervated) fed a Control or (-) L diet for 3 days. Data are expressed as the mean ± SEM (*n* represents number of samples and are indicated above the bar graph), with individual data points. Data were analyzed by two-tailed unpaired Student's *t* test for **a–c**, or one-way ANOVA followed by the SNK (Student–Newman–Keuls) test for **d–j**. Source data are provided as a Source data file.

pharmacologic ablation, which has been shown to abolish cold or POMC activation-induced browning of WAT[3,15], as well as the stimulation by β3-adrenergic agonist CL316, 243[14] in mice with inhibited amygdalar PKC-δ neuronal activity.

However, several issues remain unsolved in this study. One such question is the neuronal circuit of amygdalar GCN2 in the regulation. Previous studies have shown that essential amino acid depletion is first sensed by the anterior piriform cortex (APC) via GCN2 signals[44], the output cells of which project to hypothalamus[45]. In this study, we demonstrated an important role of GCN2 in the amygdala in the control of WAT browning under leucine deprivation. As the amygdalar neurons can also project to the hypothalamus[46], GCN2 signals in the amygdala may also function by impacting specific hypothalamic neurons under leucine deficiency. This possibility requires additional investigation.

Furthermore, it was interesting to note that overexpression of ATF4 in PKC-δ neurons promoted fat loss via WAT browning in mice exposed to chow diet, in contrast with previous studies which had shown that ATF4 deletion in AgRP neurons or POMC neurons decreases fat mass[42,47]. These disparate results suggest that ATF4 may play different roles in different neurons. In the hypothalamus, ATF4 probably functions as an important ER stress regulator, since hypothalamic ER stress has been shown to play an important role in energy homeostasis[48,49]; while in the

amygdala, because ER stress was not significantly induced, as reflected in ER stress-related changes (Supplementary Fig. 18), ATF4 most likely functions as an amino acid sensor under leucine deprivation. Therefore, overexpression of ATF4 in PKC-δ neurons probably mimics leucine deprivation-induced WAT browning.

In addition, signals mediating the activated PKC-δ neurons to the SNS, remain unclear in light of our study's results. Our results showed that deletion of GCN2 in the amygdala stimulated the activity of PKC-δ neurons, which might be mediated by increasing ATF4 expression which induces neuronal firing, as previously shown[50,51]. PKC-δ neurons are a class of GABAergic neurons, and activation of PKC-δ neurons increases GABA release thus inhibiting feeding[21]. Some PKC-δ neurons also release dopamine and corticotropin-releasing factor[52]. We suspected that the activated PKC-δ neurons may activate the SNS by releasing a type of neuropeptide. These possibilities warrant further research.

Moreover, our study failed to shed light on the relative contribution of WAT browning to fat loss under leucine deprivation. We found that body weight and fat mass reduction by leucine deprivation were prevented by deletion of amygdalar GCN2, which was associated with a blocking effect on WAT browning, oxygen consumption, and energy expenditure. These results

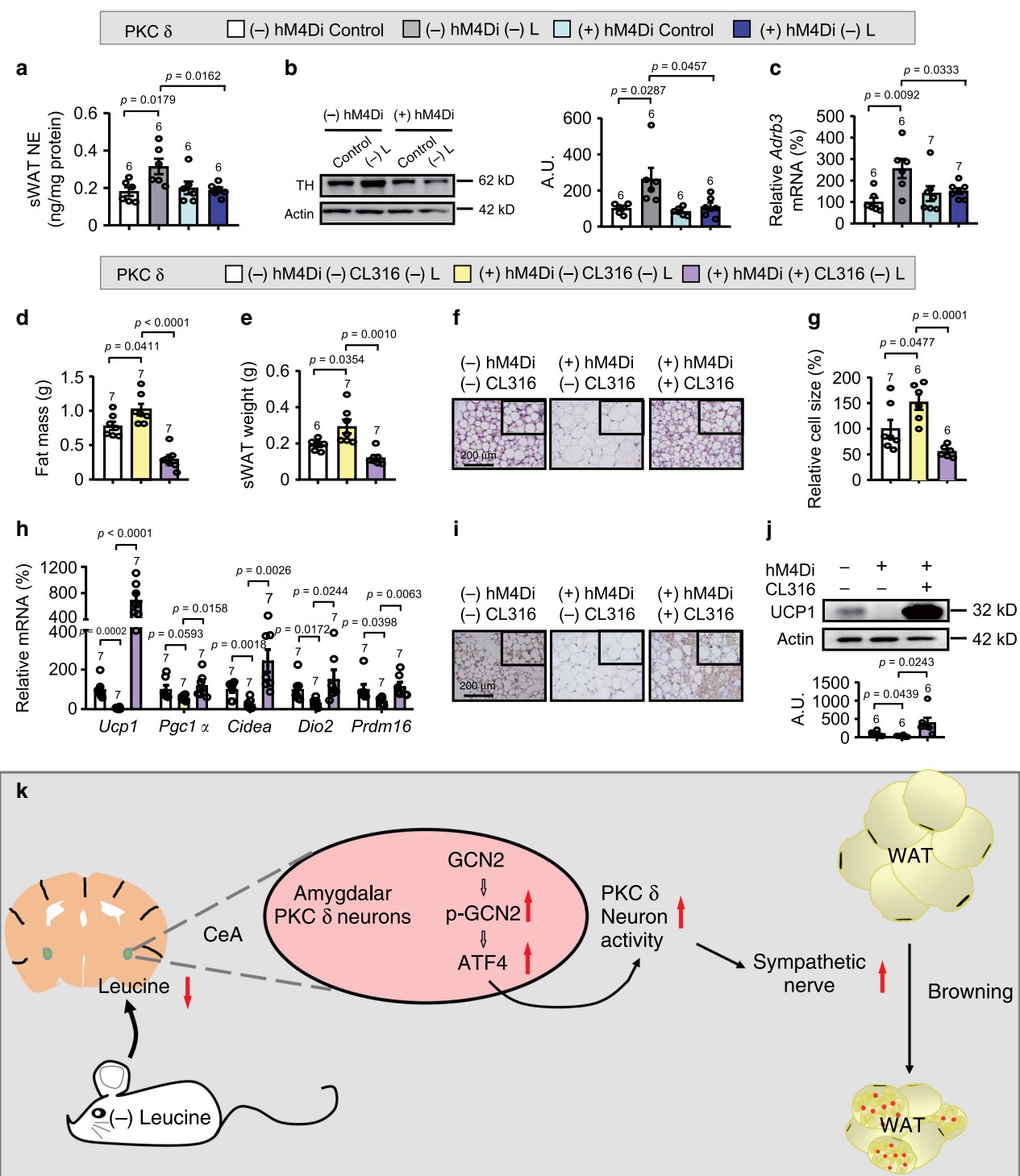

**Fig. 7 SNS activation reverses the effects of PKC-δ-neurons' inhibition on WAT browning under leucine deprivation. a** Norepinephrine (NE) levels in subcutaneous WAT (sWAT) measured by ELISA kit. **b** Tyrosine hydroxylase (TH) proteins in sWAT by western blotting (left) and quantified by densitometric analysis (right); A.U.: arbitrary units. **c** Gene expression of *Adrb3* in sWAT by RT-PCR. **d** Fat mass by NMR. **e** sWAT weight. **f** Representative images of hematoxylin and eosin (H&E) staining of sWAT. **g** sWAT cell size quantified by Image J analysis of H&E images. **h** Gene expression of *Ucp1, Pgc1a, Cidea, Dio2*, and *Prdm16* in sWAT by RT-PCR. **i** Representative images of immunohistochemistry (IHC) of UCP1 in sWAT. **j** UCP1 protein in sWAT by western blotting (top) and quantified by densitometric analysis (bottom). **k** Summary Diagram: WAT browning is induced by leucine deprivation, which is sensed by the amino acid sensor GCN2 in PKC-δ neurons of the amygdala. Activated GCN2 then subsequently stimulates ATF4 expression and increases the PKC-δ neuronal activity that promotes WAT browning via increasing the activity of the sympathetic nerve. Studies for **a–c** were conducted using 22- to 24-week-old male PKC-δ-Cre mice receiving AAVs expressing mCherry (PKC δ − hM4Di) or hM4Di (PKC δ + hM4Di), fed a control (Control) or leucine-deficient [(-) L] diet for 3 days; studies for **d–j** were conducted using 16- to 18-week-old male PKC δ − hM4Di or PKC δ + hM4Di mice injected with saline (−CL316) or CL316243 (+CL316), fed a (-) L diet for 3 days. Data are expressed as the mean ± SEM (*n* represents number of samples and are indicated above the bar graph), with individual data points. Data were analyzed by one-way ANOVA followed by the SNK (Student–Newman–Keuls) test. Source data are provided as a Source data file.

suggest that the increased WAT browning is likely to play a role in leucine deprivation-stimulated body-weight reduction and fat loss. However, other factors could not be excluded. For example, leucine deprivation-induced changes in lipolysis related gene and proteins in WAT were also blocked by deletion of amygdalar GCN2. Furthermore, we also found that reduced food intake by leucine deprivation was partly blocked in several types of mice in response to different treatments, pointing to the role of CeA neurons (including PKC-δ neurons) in food intake[21,53], also suggest that food intake might be involved in this regulation. However, the influence from the altered food intake to WAT browning under leucine deprivation was unlikely to be significant, as pair-feeding experiments showed that the reduced food intake (about 20% reduction for 3 days) had no obvious impact on the WAT browning (Supplementary Fig. 19). Moreover, though UCP1 expression in BAT was not affected by GCN2 deletion in the amygdala, thermogenesis in BAT has not been assessed. Interestingly, a differential regulation of certain specific neurons on WAT browning and BAT thermogenesis has also been reported by other studies[14,26]. Therefore, the relative contribution of WAT browning compared with other factors involved in fat loss under leucine deprivation remains to be studied.

One more issue remains unanswered is the physiological significance of this study. Previous studies[11] and those of our results have shown that leucine deprivation induces BAT UCP1 expression, which stimulates thermogenesis and helps increase the body temperature[1,54]. Because WAT browning is also considered to increase thermogenesis[2,3], we speculated that this change might help to increase thermogenesis under leucine deprivation. Therefore, it is possible that leucine deprivation causes a cold-like response that requires thermogenesis to be stimulated in BAT and WAT. These possibilities, however, require to be studied in the future.

Taken together, these results demonstrate a critical role for GCN2/ATF4 signals in amygdalar PKC-δ neurons in regulating leucine deprivation-induced WAT browning, as shown in the summary diagram (Fig. 7k). Our findings show that GCN2/ATF4 is involved in the neuronal control of the browning, and also point to an important role of the amygdala in energy homeostasis. In addition, our results suggest a way to stimulate WAT browning by manipulating nutrient intake. These results will certainly help with understanding the pathological mechanisms underlying obesity and pave the way toward the development of possible treatment for obesity.

## Methods
**Mice and diets**. All mice were of a C57BL/6J background. GCN2-floxed (GCN2$^{+/+}$) mice, Ai9 (tdTomato) reporter mice,and global GCN2 knockout mice (GCN2$^{-/-}$) were obtained from the Jackson Laboratory (Bar Harbor, ME), and PKC-δ-Cre mice were kindly provided by Prof. Yangang Sun from the Institute of Neuroscience, Chinese Academy of Sciences. To visualize PKC-δ protein-expressing neurons under the fluorescence microscope, PKC-δ-Cre mice were intercrossed with Ai9 (tdTomato) reporter mice[55]. Control (nutritionally complete amino acids) and (-) L (leucine-deficient) diets were obtained from Research Diets (New Brunswick, NJ). All diets were isocaloric and compositionally the same in terms of carbohydrate and lipid components. For the feeding experiments, mice were acclimated to a control diet for 3 days and then randomly divided into either a control or (-) L diet group with free access to either diet for 3 days or as indicated. Pair-feeding experiments were performed with wild-type (WT) mice fed a control, leucine-deficient, or pair-fed control diet for 3 days, as previously described[56]. For metabolic phenotypes over the 3-day period under leucine deprivation, WT mice were provided with a control or leucine-deficient diet for 3 days, and the food intake, body weight, fat mass, and lean mass were measured prior to and continuously over the course of the 3-day period; to assess the other parameters, another group of WT mice was provided with a leucine-deficient diet for 3 days, 2 days, 1 day, or not exposed to this diet at all (as a control), prior to being sacrificed. Mice were housed according to a 12-h light (7 a.m.)/dark (7 p.m.) cycle at 25 °C, with ad libitum access to water and rodent standard chow diet prior to the experiments. Food intake and body weight were recorded daily. These experiments were conducted in accordance with the guidelines of the Institutional

Animal Care and Use Committee of the Shanghai Institute of Nutrition and Health, Chinese Academy of Sciences.

**Ethics statements**. Animal research was complied with all relevant ethnic regulations. These studies received ethical approval by the Institutional Animal Care and Use Committee of the Shanghai Institute of Nutrition and Health, Chinese Academy of Sciences.

**Stereotaxic surgery and viral injections**. Surgery was performed with a stereotaxic frame (Steolting, IL, USA) and body temperature of the animals was maintained with a heating pad. Ophthalmic ointment was applied to maintain eye lubrication. Viruses were injected at a rate of 50 nL/min using a micro syringe pump by connecting to glass pipettes. Viral injection coordinates (in mm, midline, bregma, dorsal surface) are as following: for amygdala (±3.10, −1.42, −4.8) and for CeA (±3.00, −1.42, −4.8), in a similar strategy as shown previously[57]. After injection, the glass pipettes were left in the place for 8 min before withdrawal to allow for diffusion. The mice were allowed to recover from anesthesia on a heat blanket and then were injected with antibiotics (Ceftriaxone sodium, 0.1 g/kg) intraperitoneally for 3 days to prevent infection. Mice were individually housed and allowed to recover, while allowing time for the virus to be expressed, for at least 3–4 weeks after the surgery.

For the GCN2 ablation study, GCN2 loxp/loxp (GCN2$^{+/+}$) mice were bilaterally injected into the amygdala with an AAV vector containing a cassette expressing Cre recombinase protein with GFP (AAV9-CAG-Cre-GFP; $3.5 \times 10^{12}$ Pfu/mL) at a volume of 250 nL for each side, or an AAV vector containing a cassette expressing GFP protein (AAV9-CAG-GFP; $3.5 \times 10^{12}$ Pfu/mL) in the same volume as the control.

To knock down of GCN2 in amygdalar PKC-δ neurons, PKC-δ-Cre mice were bilaterally injected with a Cre-dependent AAV vector containing the mir-30-shGCN2 coding sequence and GFP protein in the opposite orientation flanked by two inverted loxP sites (AAV9-CMV-bGiobin-FLEX-mir-30-shGCN2-GFP; $4.9 \times 10^{12}$ Pfu/mL) at a volume of 150 nL into the CeA, or an AAV vector containing the mir-30-scramble and GFP protein in the opposite orientation flanked by two inverted loxP sites (AAV9-CMV-bGiobin-FLEX-mir-30-scramble-GFP; dilute to $4.9 \times 10^{12}$ Pfu/mL) as a control. The target sequence 5′-TCTGGATGGATTAGCTTATA-3′ for GCN2 had previously been validated[58].

To inhibit ATF4 in amygdalar PKC-δ neurons, PKC-δ-Cre mice were bilaterally injected with a Cre-dependent AAV vector containing the dominant-negative form of ATF4 (DN ATF4), with 6 amino acids mutated ($^{292}$RYRQKKR$^{298}$ to $^{292}$GYLEAAA$^{298}$) within the DNA-binding domain, as previously validated[42] in the opposite orientation flanked by two inverted loxP sites (AAV9-EF1a-DIO-DN-ATF4-mCherry, $1.3 \times 10^{12}$ Pfu/mL) at a volume of 150 nL into the CeA, or an AAV vector containing only mCherry in the opposite orientation flanked by two inverted loxP sites (AAV9-EF1a-DIO-mCherry, $1.3 \times 10^{12}$ Pfu/mL) as a control.

For over-expressing ATF4 in PKC-δ neurons, PKC-δ-Cre mice were bilaterally injected with a Cre-dependent AAV vector containing ATF4 in the opposite orientation flanked by two inverted loxP sites (AAV9-EF1a-DIO-ATF4-mCherry, $2.5 \times 10^{12}$ Pfu/mL) at a volume of 150 nL into the CeA, or an AAV vector containing only mCherry in the opposite orientation flanked by two inverted loxP sites (AAV9-EF1a-DIO-mCherry, $2.5 \times 10^{12}$ Pfu/mL) as a control.

To assess whether the activation of ATF4 in PKC-δ neurons could reverse the blocking effect of GCN2 knockdown on leucine deprivation-induced WAT browning, we injected the CeA of PKC-δ-Cre mice with Cre-dependent AAVs expressing shGCN2 (AAV9-CMV-bGiobin-FLEX-mir-30-shGCN2-GFP), ATF4 (AAV9-EF1a-DIO-ATF4-mCherry), as well as control AVVs expressing GFP (AAV9-CMV-bGiobin-FLEX-mir-30-scramble-GFP) or mCherry (AAV9-EF1a-DIO-ATF4-mCherry), as indicated, after which the mice were fed a control or leucine-deficient diet. The above AAVs were all diluted to a concentration of $2.5 \times 10^{12}$ Pfu/mL, and the two indicated AAVs were mixed at a 1:1 ratio to yield a total volume of 150 nL.

**Designer receptors exclusively activated by designer drugs (DREADDs)**. To inhibit the neuronal activity of the amygdalar PKC-δ neurons with DREADDs, PKC-δ-Cre mice were stereotaxically injected with a Cre-dependent AAV encoding an inhibitory DREADD GPCR (hM4Di) (AAV9-EF1a-DIO-hM4Di-mCherry, $3.1 \times 10^{12}$ Pfu/mL), or an AAV encoding only mCherry (AAV9-EF1a-DIO-mCherry, $3.1 \times 10^{12}$ Pfu/mL) as control, at a volume of 200 nL, bilaterally into the CeA. To activate the neuronal activity of the amygdalar PKC-δ neurons with DREADDs, PKC-δ-Cre mice were stereotaxically injected with a Cre-dependent AAV encoding an excitatory DREADD GPCR (hM3Dq) (AAV9-EF1a-DIO-hM3Dq-mCherry, $2.6 \times 10^{12}$ Pfu/mL), or an AAV encoding only mCherry (AAV9-EF1a-DIO- mCherry, $2.6 \times 10^{12}$ Pfu/mL) as control, at a volume of 200 nL bilaterally into the CeA. Four weeks after AAV delivery, all mice received intraperitoneal (i.p.) injections of CNO (MedChemExpress, NJ, USA) at 5 mg/kg of body weight for hM4Di silencing or 2 mg/kg for hM3Dq activation every 12 h for 3 days[21].

**Denervation and activation of sympathetic nervous system activity**. For the sympathetic denervation, mice were anesthetized and received 20 microinjections (1 μL per injection) of control vehicle or 6-hydroxydopamine (6-OHDA, 10 μg/μL

in saline, Sigma, MO, USA) throughout both inguinal fat pads, as previously described[15]. Seven days after pharmacologic ablation, the mice were used for the experiments. For the β3-adrenergic agonist CL316, 243[14] treatment, mice were intraperitoneally (i.p.) injected with saline or 1 mg/kg body weight of CL316, 243 (Merck Millipore, Frankfurter, GER) daily during leucine deprivation.

**Metabolic parameters measurements.** The body fat composition of mice was determined using a nuclear magnetic resonance system (Bruker, DE, USA). Indirect calorimetry was measured using a comprehensive lab animal monitoring system (CLAMS; Columbus Instruments, OH, USA)[11]. Rectal temperatures were measured using a rectal probe attached to a digital thermometer (Physitemp Instruments, NJ, USA). Norepinephrine (NE) levels were assessed using ELISA kits (Nanjing Jiancheng Bioengineering Institute, Nanjing, China) in accordance with the manufacturer's instructions.

**Histopathology and UCP1 immunohistochemistry (IHC).** White adipose tissue (WAT) and brown adipose tissue (BAT) were fixed in 4% paraformaldehyde (PFA) overnight. Tissues were embedded in paraffin and cut into 8-μm sections of entire block prepared. After deparaffinization and rehydration, sections were subjected to staining with hematoxylin and eosin (H&E).

For IHC, after deparaffinization and rehydration, WAT sections were subjected to antigen retrieval in citrate acid buffer (pH 6.0) at 95 °C for 20 min. Sections were blocked with 5% goat serum in 0.1 M phosphate buffer and 0.2% Triton X-100 for 1 h at room temperature and incubated overnight (4 °C) with anti-UCP1 antibody (1:500, Abcam, Cambridge, UK)[15]. After washing with PBS, sections were incubated with HRP-conjugated secondary antibodies (1:1000, Jackson ImmunoResearch, PA, USA).

**Electron microscopy (EM) analysis.** The amygdala was separated and fixed overnight at 4 °C with the fixative solution (4% paraformaldehyde, 0.1% gluter-aldehyde, 15% picric acid in phosphate buffer). After washing with ice-cold 0.1 M phosphate buffer, tissues were put in 1% osmium tetroxide for 15 min and dehydrated in an ethanol gradient. Uranyl acetate (1%) was added to 70% ethanol to enhance ultrastructural contrast. Tissues were then embedded in Durcupan, cut in an ultra microtome and collected in grids for posterior analyzes[59]. A JEOL transmission electron microscopy was used to visualize the ultrastructure of the samples (JEOL 1010; Tokyo, Japan).

**Immunofluorescence (IF) staining.** Mice were perfused transcardially with saline followed by PBS buffer containing 4% PFA. Brains were dissected and post-fixed overnight at 4 °C in 4% PFA, followed by cryoprotection in PBS containing 20% and 30% sucrose at 4 °C. Free-floating sections (25 μm) were prepared with a cryostat. Slices were blocked for 1 h at room temperature in PBST (0.3% Triton X-100) with 5% normal donkey serum, followed by incubation with primary antibodies at 4 °C overnight and secondary antibodies at room temperature for 2 h[60]. Primary antibodies used in IF experiments: anti-p-eIF2α (1:1000, Cell Signaling Technology, MA, USA), anti-GCN2 (1:500, Abcam, Cambridge, UK), anti-p-GCN2 (1:300, Biorbyt, Cambridge, UK), anti-c-Fos (1:500, Santa Cruz Biotechnology, CA, USA), and anti-ATF4 (1:250, Santa Cruz Biotechnology, CA, USA).

**Serum and amygdalar amino acids measurement.** Amygadala tissues were separated and homogenized with 0.1 N HCl on ice[61]. Then the tissues were centrifuged at $4500 \times g$ for 5 min and the supernatant was evaporated to dryness. To the residue, 20 μL 0.1 N HCl was added and then mixed for measurement. Serum and amygdalar amino acids were analyzed with high performance liquid chromatography (Ultimate 3000, USA) tandem mass spectrometry (API 3200 Q-TRAP, USA) methods by Beijing MS Medical Research Co. Ltd (Beijing, China).

**RNA isolation and relative quantitative RT-PCR.** RNA was extracted using TRIzol reagent (Invitrogen, CA, USA). mRNA was reverse transcribed using a High-Capacity cDNA Reverse Transcription Kit (Thermo Scientific, CA, USA) and processed for quantitative real-time PCR using SYBR Green I Master Mix reagent by ABI 7900 system (Applied Biosystems, CA, USA)[25]. The primer sequences used in this study are described in Supplementary Table 1.

**Western blotting analysis.** Tissues were homogenized in ice-cold lysis buffer (50 mM Tris HCl, pH 7.5, 0.5% Nonidet P-40, 150 mM NaCl, 2 mM EGTA, 1 mM Na$_3$VO$_4$, 100 mM NaF, 10 mM Na$_4$P$_2$O$_7$, 1 mM phenylmethylsulfonyl fluoride, 10 μg/ml aprotinin, 10 μg/ml leupeptin)[11]. Tissue extracts were then immuno-blotted with the following primary antibodies: anti-p-eIF2α, anti-EIF2α, anti-GCN2, anti-p-HSL, anti-HSL, anti-IRE1α, and anti-p-PKA substrates (1:1000, Cell Signaling Technology, MA, USA); anti-UCP1 and anti-ATF6 (1:1000, Abcam, Cambridge, UK); anti-TH (1:1000, Merck Millipore, Frankfurter, GER); anti-ATF4 and anti-TRB3 (1:500, Santa Cruz Biotechnology, CA, USA), anti-p-GCN2 (1:500, Biorbyt, Cambridge, UK); anti-CHOP, anti-ATF4, anti-PERK, and anti-XBP1s (1:1000, Proteintech, Hubei, P.R.C.); anti-p-IRE1α (1:1000, Epitomics, CA, USA); anti-p-PERK (1:1000, Signalway Antibody, MD, USA); anti-BIP and anti-β-actin (1:2000, Sigma, MO, USA).

**Statistical analysis and reproducibility.** Statistical analyses were performed in GraphPad Prism, version 8.0 (GraphPad Software, San Diego, CA). All values are presented as the mean ± standard error of the mean (SEM). When two groups were compared, data were analyzed for statistical significance using a two-tailed unpaired Student's $t$ test as indicated in the figure legends. For experiments involving multiple comparisons, data were analyzed for statistical significance using a one-way analysis of variance (ANOVA) followed by the SNK (Student–Newman–Keuls) test. In addition, the individual data points on every histogram were also shown in order to reflect the individual variability of the measures. $P$-values are indicated within the graphs. Statistical significance was defined as $p < 0.05$.

**Reporting summary.** Further information on research design is available in the Nature Research Reporting Summary linked to this article.

## Data availability

The authors declare that all data presented in this study are available within the Figures and its Supplementary Information file. The source data underlying Figs. 1–7 and Supplementary Figs. 1–19 are provided as a Source data file. Other data that support the study are available from the corresponding author upon reasonable request. Source data are provided with this paper.

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

## Acknowledgements

This work was supported by grants from the National Key R&D Program of China (2018YFA0800600), the National Natural Science Foundation (91957207, 31830044, 81870592, 81770852, 81700761, 81700750, 81970742, 81970731, 81570777, and 81600623), Basic Research Project of Shanghai Science and Technology Commission (16JC1404900 and 17XD1404200), and CAS Interdisciplinary Innovation Team, Novo Nordisk-Chinese Academy of Sciences Research Fund (NNCAS-2008-10). F.G. was also supported by the One Hundred Talents Program of CAS and Shanghai Academic Research Leader.

## Author contributions

F.G. and F.Y. planned and supervised the experimental work and data analysis; F.Y. performed the experiments and wrote the paper; H.J., H.Y., X.J., and F.J. researched data and provided technical support; S.C., H.Y., Y.C., and Q.Z. provided research material; F.G. directed the project, contributed to discussion and wrote, reviewed, and edited the paper. The paper was revised and approved by all authors.

## Competing interests

The authors declare no competing interests.
