## [Peer Review File · Nature Communications]

Reviewers' Comments:

Reviewer #1:

Remarks to the Author:

This is a very detailed and systematic analysis of the signaling between the amygdala and WAT. Given all the data that is there, it may be disturbing that I mainly will point to what is not there. Although evidently all mechanisms are of interest, we would normally consider that the outcomes of the studies should also be of physiological interest. The authors have here fully concentrated on the effects of their manipulations on the WAT browning phenomenon, with two tissues studied: sWAT and eWAT. Still, they quote from the start papers that point to BAT as the localization of the (thermogenic) effect of leucine deprivation. Why would the authors undertake all this central brain effort and not include BAT in the output? After all, the thermogenic power of sWAT is low as compared to that of BAT – and that of eWAT even lower (the UCP1 gene expression levels are often orders of magnitude different). So, did the authors also investigate BAT? If not, why not? Or did the BAT results not fit? Or do the authors save the BAT results for another paper?

This takes us to another point of apparently missing data (at least I could not find them): the authors write in the methods that they have performed indirect calorimetry and body temperature measurement – but where are the data? Do the authors think that the changes in WAT browning they report are those that explain changes in thermogenesis? How did thermogenesis fit into the regulatory picture for WAT browning?

My main problem is thus that I think that the most important physiological aspects of the study have been missed. If the authors have the BAT data and the thermogenesis data, they certainly should be included here. In the absence of this, I still find the data technically convincing, and several advanced techniques and interesting mouse strains have been utilized. But my enthusiasm is, as understood, somewhat limited.

Just a few additional points

- the paper would gain by a summarizing drawing, where all pathways investigated were placed in relation.
- there are problems with missing axes in several figures (e.g. S1D) (I can guess).
- true browning data (that must include UCP1 gene expression minimally) are not shown for S3, S4 and S5, although the text says it is.
- it is not reasonable to suggest leucine deficiency as a therapeutic tool, considering the many negative health effects of leucine deficiency
- no reference to important papers: Gettys et al. Cell Rep 2016 16, 707 and Wanders et al. Diabetes 2016 65, 1499

Reviewer #2:

Remarks to the Author:

This manuscript, describing a huge amount of work, shows that the activation of the GCN2/ATF4 pathway in amygdala PKC- δ neurons promotes WAT browning. This is an interesting and novel result providing new insights into neuronal control of WAT browning. However, some points need to be clarified to be fully enthusiastic.

General comments:

- The nutritional model: the authors have chosen to activate GCN2 by feeding animals on a -Leu diet for a long period (3days), which is a pretty drastic nutritional situation (and not very common). Long term feeding a Leu deficient diet generates a systemic catabolic state that can affect directly or indirectly several tissues. The authors should show (at least in Sup Data), the kinetic of evolution of the main biologicals parameters over the 3 days period in both control and -Leu group (at least 0, 1, 2, 3 days). I mean: body weight, food intake, fat/lean mass... Similarly, several parameters related to the GCN2/ATF4 pathway (eIF2 α -phosphorylation, ATF4 expression) and/or WAT browning should be analyzed over the 3 days period. Indeed, at least in

culture cells, eIF2a phosphorylation and ATF4 expression could be transient (at least the magnitude of the activation varies over a long period of stimulation) because negative feed-back mechanisms are turned on, it is very likely that similar mechanisms occur in vivo.

- I have also few interrogations about the animal models used for these experiments: (1) As a first experiment it would have been simple and robust to use GCN2^{-/-} mice. Is WAT browning prevented in these mice? In addition, this model would have provided robust control to study eIF2a signaling in amygdala. (2) To study the effects of a loss of function of GCN2 or ATF4 the authors expressed shRNA or a Dominant Negative form in a specific brain area or specific neurons. These technologies work but often lead to a partial loss of the target protein or a not complete inhibition of the protein function and thus could minor the result. The authors own the genetic tools to perform GCN2 or ATF4 KO specifically in PKC- δ neurons (GCN2 or ATF4-lox mice; mice strain expressing cre specifically in the PKC- δ neurons or AAV expressing cre driven by a PKC- δ specific promoter). Why the authors did not perform GCN2 or ATF4 KO specifically in PKC- δ neurons? They should justify the use of shRNA and DN technology.

-The authors clearly show that ATF4 plays a role in the regulation of WAT browning. ATF4 being downstream the pathway, its expression is induced following the activation of any of the four eIF2aKinases. Particularly, ATF4 could be induced by PERK activation resulting from ER stress. The authors do not investigate the effect of ER stress on WAT browning and do not discuss its putative role. It would be interesting to know whether ER Stress regulate WAT browning. It worth mentioned that ER Stress can be activated in brain by several nutritional or pathological situations.

-Statistical analysis could be given with more detail. Particularly, the number of mice (n) per experiment should be more clearly given.

-The manuscript is quite difficult to read. The authors could at least give more experimental details in the legend of the figures and improve the annotation of the figures (especially SD).

Point by point comments

-Figure 2A: From the western blot shown in this figure it is difficult to conclude about the magnitude of the activation of the pathway (3 days stimulation of the pathway). I suggest to measure also the expression of a few ATF4-dependent genes and to compare with GCN2^{-/-} mice.

-Figure 2BC shows that GCN2 KD in the amygdala prevents loss of fat due to -Leu diet feeding suggesting that GCN2 KD in the amygdala regulates lipolysis. Even this parameter could be different from browning it could be more deeply commented.

-Figure S6 : Additional controls should be given: Groups without shGCN2 (control group) and animals fed on a control diet should be given (to measure the possible effects of shGCN2 and/or ATF4 overexpression independently on the -leu diet).

-In discussion It is written: "In the hypothalamus, ATF4 probably is an important ER stress regulator; while in the amygdala, ATF4 mostly likely functions as an amino acid sensor. » Could you please give more details.

Reviewer #3:

Remarks to the Author:

In this study, the authors identify an interesting new role of the PKC δ neurons of the central amygdala, and identify intracellular effectors including amino sensor GCN2 and transcription factor ATF4. While very interesting and performed thoroughly, the presentation of the results is

not well organized, clarity could be improved, and the statistical analysis needs to be revised. Also, the manuscript is filled with English grammatical errors (I listed a sample in the minor comments). The text needs to be proofread by a native English speaker and/or a scientific writer.

First, as the main finding of the study is the role of the role of PKCd in WAT browning under Leucine deprivation, it would be relevant to present the experiments reporting the role of this neural populations in this metabolic response, prior to describing the experiments testing the role of intracellular effectors (GCN2 and ATF4).

Second, the statistical analysis is not appropriate. The author use multiple Student t-tests without correcting for multiple comparisons, and should consider performing ANOVA rather than t-tests. Overall, the individual data points on every histogram should also be represented to represent the individual variability of their measures.

Finally, a summary diagram/schematic would be extremely valuable for the accessibility of the manuscript. (Leucine deprivation  PKCd neurons in CeA[GCN2  eIF2a  ATF4]  ...  SNS  WAT browning)

Minor Comments

As Leucine biochemical symbol is L, the author could use this abbreviating in the manuscript: (–)L rather than (–)Leu.

Multiple abbreviations are not defined: IF (immunofluorescence), H&E (Haemotoxylin and Eosin). Similarly, GCN2 and ATF4 are used in the abstract and not defined there. The role of ATF4 is even not described.

Figure 5A: Please add the unit to the graph (%)

Figure 7: Please correct the legend. The grey and black squares on top of the figure seem to be swapped.

Figure 7H: Please correct the y axis numbers.

Line 11-13: Please rephrase, the sentence is hardly understandable

Line 16: "...of PKC-d neuron and..." Please specify "...of PKC-d neuron in the central amygdala and..."

Line 44 "on serine 51" is unnecessary, please remove

Line 45: please detail the effectors of the adaptive responses.

Line 47: regulating the other ◊ regulating other

Line 48: delete "if there is the possibility"

Line 50: replace ways with a more scientific term, such as effectors

Line 51: please cite some external stimuli

Line 56: three parts = three main parts. Indeed the amygdala also includes the basomedial and cortical amygdala.

Line 58: please specify some of the cognitive function of the amygdala.

Line 60: mediate the ◊ mediate

Line 63: our current ◊ this

Line 153 "the GCN2/eIF2a" Please be cleared. Do the author refer to the pathway ? Do these two proteins form a complex ?

Line 155: the sentence is grammatically wrong. Please replace the comma with a period. (" We speculate...")

Line 156: as downstream what ? effector ?

Lin 162, 173, 214: please replace "asked" with "tested". The authors did not only asked, they performed experiments the experiments to test the hypotheses.

Line 197: do the authors mean alteration rather than alternation ?

Line 203: The authors inappropriately mention they pharmacologically inhibit, when they chemogenetically inhibit. Please correct accordingly.

Line 215: please correct "activated the activity" with "increased the activity"...

Line 226: "in THE brain"

Line 234: pharmacologic = pharmacotoxic

Line 273: amino acid = amino acids

Line 295: mostly likely = most likely

Reviewer #4:

Remarks to the Author:

In this manuscript, the authors focused on the metabolic effects of leucine deprived diet (LDD) and revealed an interesting and important neural mechanism of LDD-induced white adipose browning. By combining multiple viral and mouse genetic tools and chemogenetic DREADDs, the authors demonstrated that LDD induced white adipose browning and adipose loss in mice. They further observed that GCN2 activation in the amygdala PKC- δ neurons and GCN2-engaged ATF4 signaling are both necessary and required to mediate LDD's such effects. Overall, the results and methods in the current manuscript are novel and findings are significant. My comments are listed below:

1. According to prior studies, LDD negatively regulates body weight by reducing food intake and increasing BAT energy expenditure. Here the authors showed further that WAT-browning is also involved. How much the effect on body weight loss from LDD is actually mediated by WAT-browning, compared with that from feeding inhibition and BAT activation? Is browning a significant component for LDD to regulate BW? The authors showed data of reduced fat mass but whether such fat mass loss is indeed mediated by WAT-browning is not clear. In addition, CeA neurons and the PKC- δ neurons in the brain region were both shown to regulate feeding and cause anorexia. When the authors manipulated these neurons by deleting/knocking down GCN2, DN- or overexpression of ATF4, and by excitatory and inhibitory DREADDs, did feeding behaviors also change? If yes, how the authors would distinguish the physiologic effects from CeA-mediated anorexia vs. browning? If no, how would the authors explain that LDD-induced fat mass loss was completely blocked in CeA-GCN2 KO/KD, DN-ATF4, and in hM4Di related studies?

2. Fig S1G/S2D: LDD induced cFos in both CeA and BLA and it seems that BLA had more notable c-Fos expression, together with stronger expression of GCN2. How BLA is involved? Since antibodies for GCN2/ATF4 worked well for IF, how many c-Fos neurons are actually GCN2+?

3. Fig S1E: Dramatic variations of Ucp1 expression were observed in the upper panel western blot results, which are obviously NOT consistent with the quantified bar data provided below.

4. Line 112-113: this sentence is confusing. I think the authors were trying to say that AAV-injected GCN2+/+ mice used as control and AAV-injected GCN2lox/lox mice (i.e. GCN2 KO mice) as the study objects. Or, AAV-GFP virus injected GCN2lox/lox mice were used as control. No matter what, a clearer description is required.

5. Fig S2B: The authors used the gene changes in the thalamus as control. However, a much better control shall be the arcuate hypothalamus, which contains important neurons expressing both GCN2 and ATF4 and previously shown to mediate both LDD's effects and WAT-browning.

6. Fig S7B, 5B-H: How CNO + LDD treatment was performed was not clear. How CNO was injected, whether saline injection of the same mice was included as control were unclear, either. More controls groups are required, particularly given the recent findings that CNO has extensive DREADDs-independent effects in the brain.

7. Many AAVs were used in the current study. However, serotypes, titer, and construction of AAVs were all missing.

8. Line 360-361: coordinates listed are confusing. Shall CeA be a part of the amygdala? Why two different coordinates were used?

9. Please improve the writing and make grammatical corrections.

Reviewers' comments:

Reviewer #1 (Remarks to the Author):

This is a very detailed and systematic analysis of the signaling between the amygdala and WAT. Given all the data that is there, it may be disturbing that I mainly will point to what is not there.

Although evidently all mechanisms are of interest, we would normally consider that the outcomes of the studies should also be of physiological interest. The authors have here fully concentrated on the effects of their manipulations on the WAT browning phenomenon, with two tissues studied: sWAT and eWAT. Still, they quote from the start papers that point to BAT as the localization of the (thermogenic) effect of leucine deprivation. Why would the authors undertake all this central brain effort and not include BAT in the output? After all, the thermogenic power of sWAT is low as compared to that of BAT - and that of eWAT even lower (the UCP1 gene expression levels are often orders of magnitude different). So, did the authors also investigate BAT? If not, why not? Or did the BAT results not fit? Or do the authors save the BAT results for another paper?

This takes us to another point of apparently missing data (at least I could not find them): the authors write in the methods that they have performed indirect calorimetry and body temperature measurement - but where are the data? Do the authors think that the changes in WAT browning they report are those that explain changes in thermogenesis? How did thermogenesis fit into the regulatory picture for WAT browning?

My main problem is thus that I think that the most important physiological aspects of the study have been missed. If the authors have the BAT data and the thermogenesis data, they certainly should be included here. In the absence of this, I still find the data technically convincing, and several advanced techniques and interesting mouse strains have been utilized. But my enthusiasm is, as understood, somewhat limited.

Our response:

1) We agree with the reviewer that the effect of amygdala on BAT should not be ignored, as BAT is the localization of thermogenic effect under leucine deprivation and the thermogenic power of sWAT and eWAT is low compared to that of BAT. In fact, we started from investigating the effect of amygdalar GCN2 on leucine deprivation-stimulated UCP1 expression in BAT at the beginning of our study. For this purpose, control (GCN2^{+/+}) mice or mice with GCN2 deletion in amygdala (GCN2 KO) were fed a control or leucine-deficient diet for 3 days. As observed in leucine deprivation for 7 days (Cheng Y et al, *Diabetes* 59(1):17-25, 2010), leucine deprivation for 3 days also significantly reduced BAT weight, increased BAT cell density as reflected in the hematoxylin and eosin (H&E) and the expression of genes (*Ucp1* and *Pgc1a*) and proteins (UCP1) related to thermogenesis regulation (Lowell

BB et al, *Nature* 404:652-660, 2010) in control mice (Fig. S1A-D). In addition, leucine deprivation increased NE levels, TH expression and *Adrb3* mRNA levels, which reflecting the activity of sympathetic nervous system (SNS) (Wang B et al, *EMBO reports* 19, 2018; Kajimura et al, *Cell metabolism* 22, 546-559, 2015), in BAT of control mice (Fig. S1E-G). Surprisingly, deletion of GCN2 in amygdala had no obvious effect on above parameters examined (Fig. S1A-G), in contrast to a significant blocking effect on WAT browning as presented in our manuscript. Consistent with our results, differential regulation of certain specific neurons on WAT browning and BAT thermogenesis has also been shown in other studies (Ruan, H. B. et al, *Cell* 159, 306-317, 2014; Wang, B. et al. *EMBO Rep* 19(4). pii: e44977, 2018; Ngoc Ly T et al, *Am J Physiol Regul Integr Comp Physiol* 312: R132–R145, 2017).

Based on the above results, we decided to concentrate on investigating the effect of amygdala on WAT browning under leucine deprivation and not include BAT data in the original manuscript. This way of presenting, however, may raise some misunderstanding, as pointed by the reviewer. Thus, we will include this part of data in the revised manuscript, as suggested. This information has been added to Results (page 7), Discussion (page 17) and Supplementary Figures (S7) in the revised manuscript.

2) Regarding the physiological aspects of WAT browning under leucine deprivation: we have performed indirect calorimetry and body temperature measurement to reflect thermogenesis in control and GCN2 KO mice under leucine deprivation and apologized for not including them in the original manuscript. We found that deletion of GCN2 in amygdala partly prevented the reduction in body weight by leucine deprivation (Fig. S2A). Then we measured the indirect calorimetry using a comprehensive lab animal monitoring system. Leucine deprivation increased oxygen consumption and energy expenditure, decreased respiratory exchange ratio (RER; V_{CO_2}/V_{O_2}), with no influence on total physical activity, in control mice (Fig. S2B-E). Though deletion of GCN2 in amygdala had no effect on RER and physical activity under leucine deprivation, it partly reduced leucine deprivation-increased oxygen consumption and energy expenditure (Fig. S2B-E). The partly reduced oxygen consumption and energy expenditure, possibly due to the blocked WAT browning, may contribute to the less reduced fat mass in GCN2 KO mice. We also measured body temperature, however, deletion of GCN2 in amygdala had no effect on leucine deprivation-increased body temperature (Fig. S2F). It is not that surprising, as the body temperature measured here referred to rectal temperature. We speculated that the effect of WAT browning may not have that big impact on the whole body temperature, but may function locally on adipose tissue thereby to increase the energy expenditure of the body. In addition, a change in thermogenesis is not always accompanied by a change in body temperature, as shown in many other studies (Shin H et al, *Cell Metab* 26(5):764-777, 2017; Zhang S et al, *PLoS Biol* 16(5):e2004225, 2018). Taken together, though we did not observe a significant change in body temperature, the partly reduced oxygen consumption and energy expenditure by GCN2 deletion in amygdala under leucine deprivation suggest that the increased

WAT browning is likely to play a role in leucine deprivation-stimulated fat loss. The possible contribution of WAT browning to leucine deprivation-induced metabolic effects, however, remains to be studied in the future.

We agree with the reviewer that the outcome of the study should be of physiological interest under leucine deprivation and have added this information to Results (page 7), Discussion (pages 16 and 17) and Supplementary Figures (S8) in the revised manuscript.

Supplementary Figure 1. Deletion of GCN2 in amygdala has no obvious effect on brown adipose tissue (BAT) under leucine deprivation.

A: BAT weight;

B: Representative images of hematoxylin and eosin (H&E) staining of BAT;

C: Gene expression of *Ucp1* and *Pgc1α* in BAT by RT-PCR;

D: UCP1 protein in BAT by western blotting (top) and quantified by densitometric analysis (bottom);

E: Norepinephrine (NE) levels in BAT measured by ELISA kit;

F: TH protein in BAT by western blotting (top) and quantified by densitometric analysis (bottom);

G: Gene expression of *Adrb3* in BAT by RT-PCR.

All studies were conducted using 20- to 22-week-old male control mice (GCN2^{+/+}) or mice with GCN2 deletion in amygdala (GCN2 KO) fed a control (Con) or leucine-deficient [(-) L] diet for 3 days. Data are expressed as the mean ± SEM (n = 6–7 mice/group, as indicated), with individual data points. Data were analyzed by one-way ANOVA followed by the SNK (Student–Newman–Keuls) test. **P* < 0.05 for the effect of any group versus control mice under a Con diet; #*P* < 0.05 for the effect

of GCN2 KO mice versus control mice both under a (-) L diet.

Supplementary Figure 2. Metabolic parameters in mice with GCN2 deletion in amygdala under leucine deprivation.

- A: Body weight change relative to original body weight;
- B: 24-h oxygen consumption normalized by lean mass measured by the comprehensive lab animal monitoring system (CLAMS);
- C: Energy expenditure (EE) measured by CLAMS;
- D: Respiratory exchange ratio (RER, V_{CO_2}/V_{O_2}) measured by CLAMS;
- E: Locomotor activity measured by CLAMS;
- F: Rectal temperature by the digital thermometer.

All studies were conducted using 20- to 22-week-old male control mice (GCN2^{+/+}) or mice with GCN2 deletion in amygdala (GCN2 KO) fed a control (Con) or leucine-deficient [(-) L] diet for 3 days. Data are expressed as the mean ± SEM (n = 4–7 mice/group, as indicated), with individual data points. Data were analyzed by one-way ANOVA followed by the SNK (Student–Newman–Keuls) test. **P* < 0.05 for the effect of any group versus control mice under a Con diet; #*P* < 0.05 for the effect of GCN2 KO mice versus control mice both under a (-) L diet.

Just a few additional points

– the paper would gain by a summarizing drawing, where all pathways investigated were placed in relation.

Our response:

We appreciate it very much for the reviewer’s suggestion and have drawn a summary diagram as shown below (Fig. S3). This information has been added to Discussion (page 17) and Figures (7K).

Supplementary Figure 3. The summary diagram of amygdalar GCN2 regulation of WAT browning under leucine deprivation.

White adipose tissue (WAT) browning is induced by leucine deprivation, which is sensed by the amino acid sensor GCN2 in PKC- δ neurons of the amygdala. Activated GCN2 then subsequently stimulates ATF4 expression and increases the PKC- δ neuronal activity that promotes WAT browning via increasing the activity of the sympathetic nerve.

- there are problems with missing axes in several figures (e.g. S1D) (I can guess).

Our response:

We appreciate it very much for the reviewer's careful reading of our manuscript. In response, we have added axes as shown below (Fig. S4) in Supplementary Figures (S1D) in the revised manuscript.

Supplementary Figure 4. Gene expression of *Ucp1*, *Pgc1a*, *Cidea*, *Dio2* and *Prdm16* in epididymal white adipose tissue by RT-PCR.

Studies were conducted using 14- to 15-week-old male wild-type mice fed a control (Con) or leucine-deficient [(-) L] diet for 3 days. Data are expressed as the mean \pm SEM (n = 6 mice/group), with individual data points. Data were analyzed by

by two-tailed unpaired Student's t test. * $P < 0.05$ for the effect of (-) L versus Con diet group.

- true browning data (that must include UCP1 gene expression minimally) are not shown for S3, S4 and S5, although the text says it is.

Our response:

We appreciate it very much for the reviewer's pointing out not showing the expression of browning markers in eWAT in Fig. S3, S4 and S5 mentioned in the original manuscript. In response to the reviewer's inquiry, the expression of *Ucp1* and other browning markers were examined in eWAT of those mice mentioned in these Figures. As observed in sWAT, the expression of these genes were increased by leucine deprivation in control mice, but were unchanged in PKC δ -shGCN2 mice or PKC δ -DN ATF4 mice (Fig. S5A-D). Meanwhile, over-expression of ATF4 in amygdalar PKC δ neurons stimulates expression of these genes in eWAT (Fig. S5E). UCP1 protein levels were also changed accordingly under each condition (Fig. S5). These results are consistent with those observed in sWAT and what we wrote in the manuscript.

This information has been added to Supplementary Figures (S9G, S9H, S12K, S12L, S13L and S13M) in the revised manuscript.

Supplementary Figure 5. Expression of browning markers in epididymal white adipose tissue (eWAT) of mice under different treatments.

A, C and E: Gene expression of *Ucp1*, *Pgc1a*, *Cidea*, *Dio2* and *Prdm16* in eWAT by RT-PCR;

B, D and F: UCP1 protein in eWAT by western blotting (top) and quantified by densitometric analysis (bottom).

Studies for A and B were conducted using 13- to 16-week-old male PKC- δ -Cre mice receiving AAVs expressing GFP (PKC δ - shGCN2) or shGCN2 (PKC δ + shGCN2) fed a control (Con) or leucine-deficient [(-) L] diet for 3 days; studies for C and D were conducted using 13- to 16-week-old male PKC- δ -Cre mice receiving AAVs expressing mCherry (PKC δ - DN ATF4) or DN ATF4 (PKC δ + DN ATF4) fed a Con or (-) L diet for 3 days; studies for E and F were conducted using or 13- to 15-week-old male PKC- δ -Cre mice receiving AAVs expressing mCherry (PKC δ - ATF4) or ATF4 (PKC δ + ATF4), fed a Con or (-) L diet for 3 days. Data are expressed as the mean \pm SEM (n = 6-8 mice/group as indicated), with individual data points. Data were analyzed by one-way ANOVA followed by the SNK (Student–Newman–Keuls) test for A-D, or by two-tailed unpaired Student's t test for E and F. * $P < 0.05$ for the effect of any group versus control mice under a Con diet; # $P < 0.05$ for the effect of PKC δ + shGCN2 mice or PKC δ + DN ATF4 mice versus control mice all under a (-) L diet.

- it is not reasonable to suggest leucine deficiency as a therapeutic tool, considering the many negative health effects of leucine deficiency

Our response:

In our study, we treated mice with leucine-deficient diet for 3 days and assumed that it is unlikely to cause any significant injury, therefore suggest it might be a new way of stimulating WAT browning and have great prospects concerning the treatment of obesity and other metabolic disorders. However, as mentioned by the reviewer, the potential negative health effects following long-time use of leucine deficiency should be highly considered. We appreciate it very much for the reviewer's pointing out this issue and have removed the description in Discussion (page 13) in the revised manuscript.

- no reference to important papers: Gettys et al. Cell Rep 2016 16, 707 and Wanders et al. Diabetes 2016 65, 1499

Our response:

We appreciate it very much for the reviewer's suggestion and also feel that these studies provide novel and very important insights into understanding the mechanisms underlying amino acid sensing. As suggested, we have added these two papers in Introduction (pages 3) in the revised manuscript.

Reviewer #2 (Remarks to the Author):

This manuscript, describing a huge amount of work, shows that the activation of the GCN2/ATF4 pathway in amygdala PKC-neurons promotes WAT browning. This is an interesting and novel result providing new insights into neuronal control of WAT browning. However, some points need to be clarified to be fully enthusiastic.

General comments:

- The nutritional model: the authors have chosen to activate GCN2 by feeding animals on a -Leu diet for a long period (3 days), which is a pretty drastic nutritional situation (and not very common). Long term feeding a Leu deficient diet generates a systemic catabolic state that can affect directly or indirectly several tissues. The authors should show (at least in Sup Data), the kinetic of evolution of the main biologicals parameters over the 3 days period in both control and -Leu group (at least 0, 1, 2, 3 days). I mean: body weight, food intake, fat/lean mass...

Similarly, several parameters related to the GCN2/ATF4 pathway (eIF2a-phosphorylation, ATF4 expression) and/or WAT browning should be analyzed over the 3 days period. Indeed, at least in culture cells, eIF2a phosphorylation and ATF4 expression could be transient (at least the magnitude of the activation varies over a long period of stimulation) because negative feed-back mechanisms are turned on, it is very likely that similar mechanisms occur in vivo.

Our response:

We agree with the reviewer that long-term feeding a leucine-deficient diet may generate a systemic catabolic state that can affect directly or indirectly several tissues. In response, we examined the kinetic of evolution of some of the main parameters over the three days period under leucine deprivation. Wild-type (WT) mice were provided with a control or leucine-deficient diet for three days; and the food intake, body weight, fat mass and lean mass were measured prior to and continuously over the three days period. The food intake was decreased from the first day after leucine deficiency and continuously over the three days period (Fig. S6A). The body weight, fat mass and lean mass measured by NMR, showed decreased tendency one day after leucine deprivation, but was significantly decreased two days after leucine deprivation and lasted for the three days period of leucine deficiency (Fig. S6B-D). For analyzing other parameters, another group of WT mice were provided with a leucine-deficient diet three days, two days, one day or without this diet (as a control) prior to be sacrificed. The weight of sWAT and eWAT showed decreased tendency one or two day after leucine deprivation, but was only obviously decreased after three days leucine deficiency (Fig. S6E). H&E staining also showed gradually decreased lipid droplets in sWAT and eWAT following leucine deprivation (Fig. S6F). We then examined expression of WAT browning markers including *Ucp1*, *Pgc1a*, *Cidea*, *Dio2* and *Prdm* and found that they were increased from the first day after leucine deficiency and lasted for the three days period, with UCP1 protein levels changed

accordingly (Fig. S6G-J).

We then analyzed parameters related to GCN2/ATF4 pathway in amygdala over the three days period of leucine deprivation. The amygdalar p-GCN2 and p-eIF2 α levels were increased from the first day after leucine deficiency and lasted for three days during the period under leucine deprivation (Fig. S6K). The expression of ATF4 and its downstream target TRB3 (Zhang, Q. et al, *Diabetes* 62, 2230-2239, 2013), also showed increased tendency from the first day of leucine deprivation, but were increased significantly after two days leucine deprivation (Fig. S6K).

These results show that the effect of leucine deprivation on WAT browning and signals regulating this process started after the first day of leucine deprivation and lasted for the whole period for experiment, whereas most of the other parameters showed decreased tendency and became obvious after three day's leucine deprivation as a result. This information has been added to Results (page 4) and Supplementary Figures (S2) in the revised manuscript.

Supplementary Figure 6. Metabolic parameters over the three days period under leucine deprivation.

- A: Food intake;
- B: Body weight;
- C: Fat mass by NMR;
- D: Lean mass by NMR;
- E: Subcutaneous white adipose tissue (sWAT) and epididymal WAT (eWAT) weight;
- F: Representative images of hematoxylin and eosin (H&E) staining of sWAT and eWAT;
- G: Gene expression of *Ucp1*, *Pgc1α*, *Cidea*, *Dio2* and *Prdm16* in sWAT by RT-PCR;
- H: UCP1 protein in sWAT by western blotting (top) and quantified by densitometric analysis (bottom);
- I: Gene expression of *Ucp1*, *Pgc1α*, *Cidea*, *Dio2* and *Prdm16* in eWAT by RT-PCR;
- J: UCP1 protein in eWAT by western blotting (top) and quantified by densitometric analysis (bottom);
- K: p-GCN2, t-GCN2, p-eIF2α, t-eIF2α, ATF4 and TRB3 proteins in amygdala by western blotting (left) and quantified by densitometric analysis (right).

Studies were conducted using 16- to 17-week-old male wild-type mice fed a control (Con) or leucine-deficient [(-) L] diet for three days for A-D, or provided with a leucine-deficient diet one day [(-) L 1 d], two days [(-) L 2 d], three days [(-) L 3 d] or without this diet (Con) prior to be used for E-K. Data are expressed as the mean ± SEM (n = 6 mice/group), with individual data points. Data were analyzed by two-tailed unpaired Student's t test for A-D, or by one-way ANOVA followed by the SNK (Student–Newman–Keuls) test for E-K. *P < 0.05 for the effect of any group versus control mice under a Con diet.

- I have also few interrogations about the animal models used for these experiments: (1) As a first experiment it would have been simple and robust to use GCN2^{-/-} mice. Is WAT browning prevented in these mice? In addition, this model would have provided robust control to study eIF2 α signaling in amygdala.

Our response:

We agree with the reviewer that using GCN2^{-/-} mice to study its role in WAT browning during leucine deprivation would be robust and simple. In response to the reviewer's inquiry, we investigated WAT browning in wild-type (WT) and GCN2^{-/-} (KO) mice maintained on a control or leucine-deficient diet for 3 days. As predicted, leucine deprivation significantly reduced body fat mass, including subcutaneous WAT (sWAT) and epididymal WAT (eWAT), caused a distinct histological morphology with the presence of multilocular small lipid droplets and induced expression of genes (*Ucp1*, *Pgc1 α* , *Cidea*, *Dio2* and *Prdm16*) or proteins (UCP1) related to WAT browning in sWAT and eWAT of WT mice (Fig. S7A-G). These effects of leucine deprivation, however, were blocked in GCN2^{-/-} mice (Fig. S7A-G). Moreover, leucine deprivation-increased levels of p-eIF2 α in amygdala were also blocked in GCN2^{-/-} mice (Fig. S7H). These results suggest an important role of GCN2 in the regulation of WAT browning under leucine deprivation, which provide important evidence for studying GCN2 in this study.

This information has been added to Results (page 6) and Supplementary Figures (S4) in the revised manuscript.

Supplementary Figure 7. Leucine deprivation-stimulated white adipose tissue (WAT) browning is blocked in GCN2^{-/-} mice.

A: Fat mass by NMR;

B: Subcutaneous WAT (sWAT) and epididymal WAT (eWAT) weight;

C: Representative images of hematoxylin and eosin (H&E) staining of sWAT and eWAT;

D: Gene expression of *Ucp1*, *Pgc1a*, *Cidea*, *Dio2* and *Prdm16* in sWAT by RT-PCR;

E: UCP1 protein in sWAT by western blotting (top) and quantified by densitometric analysis (bottom);

F: Gene expression of *Ucp1*, *Pgc1a*, *Cidea*, *Dio2* and *Prdm16* in eWAT by RT-PCR;

G: UCP1 protein in eWAT by western blotting (top) and quantified by densitometric analysis (bottom);

H: P-eIF2α and t-eIF2α proteins in amygdala by western blotting (left) and quantified by densitometric analysis (right).

Studies were conducted using 9- to 10-week-old male wild-type (WT) or GCN2^{-/-} (KO) mice fed a control (Con) or leucine-deficient [(-) L] diet for 3 days. Data are expressed as the mean ± SEM (n = 5 mice/group), with individual data points. Data were analyzed by one-way ANOVA followed by the SNK (Student–Newman–Keuls) test. *P < 0.05 for the effect of any group versus control mice under a Con diet; #P < 0.05 for the effect of KO mice versus control mice under a (-) L diet.

(2) To study the effects of a loss of function of GCN2 or ATF4 the authors expressed shRNA or a Dominant Negative form in a specific brain area or specific neurons. These technologies work but often lead to a partial loss of the target protein or a not complete inhibition of the protein function and thus could minor the result. The authors own the genetic tools to perform GCN2 or ATF4 KO specifically in PKC⁻ neurons (GCN2 or ATF4-lox mice; mice strain expressing cre specifically in the PKC⁻ neurons or AAV expressing cre driven by a PKC⁻ specific promoter). Why the authors did not perform GCN2 or ATF4 KO specifically in PKC⁻ neurons? They should justify the use of shRNA and DN technology.

Our response:

We appreciate it very much for the reviewer's suggestion and agree that using PKC-δ Cre mice mating with GCN2^{loxp/loxp} or ATF4^{loxp/loxp} mice would be able to

fully knock out the target gene. However, since PKC- δ is not only expressed in CeA, but also in thalamus of brain and other organs like intestine in the body (Haubensak et al, *Nature* 468, 270-276, 2010; Mecklenbräuer I et al, *Nature* 416(6883):860-5, 2002; Kho DH et al, *Gut* 58(4):509-19, 2009), generating knockout mice in this way would cause GCN2 or ATF4 to be deleted in other neurons or tissues, in addition to amygdalar PKC- δ neurons. That is why we chose to knockdown GCN2 or ATF4 expression by injection of Cre-dependent AAV, the strategy that has been commonly used in studying signals in certain specific areas of brain (Ji G et al, *J Neurosci* 37(6):1378-1393, 2017; Cui Y et al, *Cell Reports* 21(7):1770-1782, 2017; Luo R et al, *Nature Communications* 9(1):2483, 2018; Xu J et al, *Nature* 556(7702):505-509, 2018). Though the method has been broadly used, we agree with the reviewer that to verify the efficiency of AAVs used would be the key step in the study.

As for construction of the Cre-dependent shGCN2 AAV, we chose the sequence that has been validated to function efficiently for inhibiting GCN2 (Maurin, A. C. et al, *Cell reports* 6, 438-444, 2014). To validate knockdown efficiency of GCN2 in PKC- δ - shGCN2 mice, we have previously shown that GCN2 was largely inhibited in these neurons by IF staining (Fig. S3C in original manuscript). However, this inhibition of GCN2 was not observed in other areas of amygdala, such as BLA (Fig. S8A). We have also conducted additional experiments by examining mRNA and protein levels of GCN2 in the amygdala of these mice and found that GCN2 expression was reduced by shGCN2 AAV in amygdala but not other brain area such as hypothalamus (Fig. S8B and 8C). The reduction of GCN2 was not complete, possibly because PKC- δ neurons accounts for only a part of amygdala, it would be difficult to observe a complete knockdown efficiency when GCN2 in other neurons was not suppressed. To further demonstrate the knockdown efficiency of shGCN2, we examined the levels of p-GCN2 and p-eIF2 α in the amygdala of mice under leucine deprivation and found that leucine deprivation-increased levels of p-GCN2 and p-eIF2 α were blocked in mice injected with AAV expressing shGCN2 (Fig. S8D). These results suggest that the AAV expressing shGCN2 used in our study was reasonable to be used for studying the role of GCN2 in amygdala.

To construct Cre-dependent AAV inhibiting the function of ATF4, we used dominant-negative form of ATF4 (DN-ATF4) with mutation of 6 amino acids (²⁹²R Y R Q K K R ²⁹⁸ to ²⁹²G Y L E A A A ²⁹⁸) within the DNA-binding domain (HE CH et al, *J Biol Chem* 276(24):20858-65, 2001). DN-ATF4 has been commonly used to suppress the function of ATF4, as reported in many studies (HE CH et al, *J Biol Chem* 276(24):20858-65, 2001; Roybal CN et al, *J Biol Chem* 279(15):14844-52, 2004; Huang H et al, *J Inflamm* 12:31, 2015). To validate the efficiency of DN-ATF4, we transfected primary amygdalar neurons with adenovirus expressing the same sequence of DN-ATF4 as for AAV or control Ad-GFP and found that the expression of TRB3, the downstream target of ATF4 (Ohoka N et al, *EMBO J* 24(6):1243-55, 2005), was significantly reduced by DN-ATF4 (Fig. S8E and 8F). Consistently, TRB3 expression was increased by over-expression of ATF4 in primary amygdalar neurons (Fig. S8G and 8H). These results suggest that TRB3 could be used as a marker reflecting the activity of ATF4. As observed for in vitro study, we also found decreased or increased

amygdalar TRB3 expression in mice with ATF4 inhibition or over-expression in PKC- δ neurons, respectively, as shown in our original manuscript. However, TRB3 expression was not affected in the hypothalamus of these mice under either case (Fig. S8I-L). Based on the changes of TRB3 and the IF staining of DN-ATF4, we believe that DN-ATF4 employed in our study should be sufficient to inhibit ATF4 and used for studying the role of ATF4 in amygdala.

In response, to demonstrate the deletion efficiency better, we have replaced Fig. S3C in original manuscript that includes CeA only with the data that includes a wider filed including both CeA and BLA (Fig. S9C) in the revised manuscript. In addition, relative information has been added to Results (pages 8 and 10) and Supplementary Figures (S12E, S12F, S13B and S13C) and in the revised manuscript.

Supplementary Figure 8. Validation of the efficiency of AAVs expressing shGCN2 or DN-ATF4.

A: Immunofluorescence (IF) staining for GFP (green), GCN2 (red) or merge (yellow) in amygdala (Amy); CeA: the central nuclei of the amygdala; BLA: the basolateral nuclei of the amygdala;

B: Gene expression of *Gcn2* in Amy and hypothalamus (Hypo) by RT-PCR;

C: GCN2 proteins in Amy and Hypo by western blotting (top) and quantified by densitometric analysis (bottom);

D: P-GCN2, t-GCN2, p-eIF2 α and t-eIF2 α proteins in Amy by western blotting (left) and quantified by densitometric analysis (right);

E: Gene expression of *Atf4* and *Trb3* in primary amygdala neurons by RT-PCR;

F: ATF4 and TRB3 proteins in primary amygdala neurons by western blotting (left) and quantified by densitometric analysis (right);

G: Gene expression of *Atf4* and *Trb3* in primary amygdala neurons by RT-PCR;

H: ATF4 and TRB3 proteins in primary amygdala neurons by western blotting (left) and quantified by densitometric analysis (right);

I and K: Gene expression of *Atf4* and *Trb3* in Hypo by RT-PCR;

J and L: ATF4 and TRB3 proteins in Hypo by western blotting (left) and quantified by densitometric analysis (right).

Studies for A-D were conducted using 13- to 16-week-old male PKC- δ -Cre mice receiving AAVs expressing GFP (PKC δ - shGCN2) or shGCN2 (PKC δ + shGCN2) fed a control (Con) or leucine-deficient [(-) L] diet for 3 days; studies for I and J were conducted using 13- to 16-week-old male PKC- δ -Cre mice receiving AAVs expressing mCherry (PKC δ - DN ATF4) or DN ATF4 (PKC δ + DN ATF4) fed a Con or (-) L diet for 3 days; studies for K and L were conducted using 13- to 15-week-old male PKC- δ -Cre mice receiving AAVs expressing mCherry (PKC δ - ATF4) or ATF4 (PKC δ + ATF4) fed a Con or (-) L diet for 3 days; studies for E and F were conducted using primary amygdala neurons prepared as previously described (Hay CW et al, *Psychoneuroendocrinology* 47:43-55, 2014) infected with adenovirus expressing GFP (- Ad-DN ATF4) or DN ATF4 (+ Ad-DN ATF4); studies for G and

H were conducted using primary amygdala neurons infected with adenovirus expressing GFP (- Ad-ATF4) or Ad-ATF4 (+ Ad-ATF4). Data are expressed as the mean \pm SEM (n = 6 mice/group for A-D and I-L; n = 8/group for E-H), with individual data points. Data were analyzed by two-tailed unpaired Student's t test for B and C, E-L; data were analyzed by one-way ANOVA followed by the SNK (Student–Newman–Keuls) test for D. * $P < 0.05$ for the effect of any group versus control mice under a Con diet for A-D and I-L, or any group versus control group for E-H; # $P < 0.05$ for the effect of PKC δ + shGCN2 mice versus control mice both under a (-) L diet for D.

-The authors clearly show that ATF4 plays a role in the regulation of WAT browning. ATF4 being downstream the pathway, its expression is induced following the activation of any of the four eIF2aKinases. Particularly, ATF4 could be induced by PERK activation resulting from ER stress. The authors do not investigate the effect of ER stress on WAT browning and do not discuss its putative role. It would be interesting to know whether ER Stress regulate WAT browning. It worth mentioned that ER Stress can be activated in brain by several nutritional or pathological situations.

Our response:

We appreciate it very much for the reviewer's pointing out this interesting possibility. Several studies have shown that ER stress is involved in the regulation of WAT browning. For example, activation of ER stress in macrophages suppresses WAT browning (Shan B et al, *Nat Immunol* 18(5):519-529, 2017) and decreased ER stress in hypothalamus is associated with or activates WAT browning (Martínez-Sánchez N et al, *Cell Metab* 26(1):212-229, 2017; Contreras C et al, *Diabetes* 66(1):87-99, 2017). Because ATF4 is a well-known regulator for ER stress induced by activation of protein kinase RNA-like ER kinase (PERK) (Balsa E et al, *Mol Cell* 74(5):877-890, 2017) and ER stress is induced by several nutritional and pathological situations (Balsa E et al, *Mol Cell* 74(5):877-890, 2017; An H et al, *Mol Cell* 74(5):891-908, 2019; Binet F et al, *Cell Metab* 22(4):560-75, 2015), it is conceivable to hypothesize that ER stress might be involved in Amygdalar ATF4 regulation of WAT browning under leucine deprivation.

To test this possibility, we examined genes and proteins related to ER stress in the amygdala of mice maintained on a control or leucine-deficient diet for three days. The genes include those inducing ER stress including PERK, activating transcription factor 6 (ATF6) and inositol-requiring transmembrane enzyme 1 (IRE1), as well as others reflecting the change of ER stress, including spliced X-box binding protein 1 (XBP1), C/EBP homologous protein (CHOP) and Bip (Hsp70) (Karali E et al, *Mol Cell* 54(4):559-72, 2014; Yoshida H et al, *Cell* 107(7):881-91, 2001; Amin-Wetzel N et al, *Cell* 171(7):1625-1637, 2017).

After leucine deprivation, *Atf4*, *Chop* and *Bip* expression were increased, with other genes unaffected, in the amygdala (Fig. S9A). We then examined protein levels of these signals related to ER stress. Except for CHOP and ATF4, the other proteins

or phosphorylated-proteins were either unchanged or decreased in the amygdala of mice maintained on a leucine-deficient diet (Fig. S9B). Then we conducted the electron microscope analysis to investigate whether there were any morphological changes in ER of the amygdala of mice under a control or leucine-deficient diet. No obvious changes reflecting ER stress, including the swelling or damaged ER, or abnormal mitochondria-ER contacts (Arruda AP et al, *Nat Med* 20(12):1427-35, 2014; Kishino A et al, *Sci Rep* 7(1):4442, 2017; Wang Y et al, *Invest Ophthalmol Vis Sci* 60(1):265-273, 2019), were observed between mice fed a control or leucine-deficient diet (Fig. S9C). These results suggest that no significant ER stress was induced in the amygdala by leucine deprivation.

Because ER stress was not significantly induced, we speculated that ER stress is unlikely to be involved in amygdalar ATF4 regulation of WAT browning. However, giving the importance of ER stress in WAT browning, we have added this information to Discussion (page 15) and Supplementary Figures (S17) in the revised manuscript.

Supplementary Figure 9. The signals of ER stress in amygdala under leucine deprivation.

A: Gene expression of *Ire1α*, *Atf4*, *Chop*, *Bip*, *Xbp1u*, *Xbp1s* and *Atf6* by RT-PCR;
B: P-PERK, t-PERK, p-IRE1α, t-IRE1α, ATF4, CHOP, BIP, XBP1s and ATF6 proteins by western blotting (top) and quantified by densitometric analysis (right);
C: Electron microscopy (EM) analysis of the amygdala; ER: endoplasmic reticulum; M: mitochondria.

Studies were conducted using 14- to 15-week-old male wild-type mice fed a control (Con) or leucine-deficient [(-) L] diet for 3 days. Data are expressed as the mean ± SEM (n = 6 mice/group), with individual data points. Data were analyzed by two-tailed unpaired Student's t test. *P < 0.05 for the effect of (-) L versus Con diet

group.

-Statistical analysis could be given with more detail. Particularly, the number of mice (n) per experiment should be more clearly given.

Our response:

We appreciate it very much for the reviewer's suggestion. We have added more details to statistical analysis in Methods (page 24), Figure Legends (pages 28-39) and Supplementary Figure Legends (pages 1-18). In addition, we have designated the number of mice per experiments by showing individual data points in all of the histograms of Figures (Fig. 1-7) and Supplementary Figures (S1-S18).

-The manuscript is quite difficult to read. The authors could at least give more experimental details in the legend of the figures and improve the annotation of the figures (especially SD).

Our response:

As suggested, we have added more experimental details to all of the Figure Legends (pages 28-39) and Supplementary Figure Legends (pages 1-28). We have also improved the annotation of all of Figures (Fig. 1-7) and Supplementary Figures (S1-S18) by providing with more details.

Point by point comments

-Figure 2A: From the western blot shown in this figure it is difficult to conclude about the magnitude of the activation of the pathway (3 days stimulation of the pathway). I suggest to measure also the expression of a few ATF4-dependent genes and to compare with GCN2^{-/-} mice.

Our response:

As pointed by the reviewer, the magnitude of p-eIF2a change may disturb the evaluation for the activation of the pathway. However, we have other data suggesting the activation of this pathway, including the change of p-GCN2 in Figure 2A and the changes of *Atf4* and *Trb3* expression in Figure 4A (in the original manuscript), in the amygdala of mice maintained on a leucine-deficient diet. To provide more convincing evidence, as suggested by the reviewer, we carried out additional experiments by examining the expression of a few ATF4-dependent genes, including *Trb3*, *activating transcription factor 3 (Atf3)* and *growth-arrest- and DNA-damage-induced transcript 34 (GADD34)*; (Wortel IMN et al, *Trends Endocrinol Metab* 28(11):794-806, 2017), in the amygdala of control mice and GCN2 KO mice maintained on a control or leucine-deficient diet. As predicted, leucine deprivation induced expression of these genes in the amygdala of control mice, but not in GCN2 KO mice (Fig. S10). These results further confirm an activation of GCN2-dependent pathway in amygdala under leucine deprivation.

This information has been added to Results (page 7) and Supplementary Figures (S5E) in the revised manuscript.

Supplementary Figure 10. GCN2 deletion in amygdala inhibits the expression of ATF4-dependent genes under leucine deprivation.

Gene expression of *Atf4*, *Trb3*, *Atf3* and *Gadd34* by RT-PCR were analyzed in the amygdala of 20- to 22-week-old male control mice (GCN2^{+/+}) or mice with GCN2 deletion in amygdala (GCN2 KO) fed a control (Con) or leucine-deficient [(-) L] diet for 3 days. Data are expressed as the mean \pm SEM (n = 6 mice/group), with individual data points. Data were analyzed by one-way ANOVA followed by the SNK (Student–Newman–Keuls) test. * $P < 0.05$ for the effect of any group versus control mice under a Con diet; # $P < 0.05$ for the effect of GCN2 KO mice versus control mice both under a (-) L diet.

-Figure 2BC shows that GCN2 KD in the amygdala prevents loss of fat due to -Leu diet feeding suggesting that GCN2 KD in the amygdala regulates lipolysis. Even this parameter could be different from browning it could be more deeply commented.

Our response:

We agree with the reviewer's hypothesis, as WAT lipolysis is also induced under leucine deprivation, which is stimulated by activation of sympathetic nervous system (SNS) (Cheng Y *et al*, *Diabetes* 59(1):17-25, 2010). In this study, we showed that knocking down of GCN2 in amygdala inhibited SNS activity, therefore, it is very likely that lipolysis in WAT was suppressed, which may contribute to the blocking effect of fat loss in these mice under leucine deprivation. To test this possibility, we examined the protein levels of the phosphorylated hormone-sensitive lipase (p-HSL), a key enzyme regulating lipolysis (Zechner R *et al*, *Cell metabolism* 15:279-291, 2012), and phosphorylated substrate for PKA, the kinase that phosphorylates HSL (Zechner R *et al*, *Cell metabolism* 15:279-291, 2012), in sWAT of mice maintained on a control or leucine-deficient diet for 3 days. As shown previously (Cheng Y *et al*, *Diabetes* 59(1):17-25, 2010), leucine deprivation increased p-HSL and p-PKA substrate in sWAT of control mice, but not in mice with GCN2 knockdown in amygdala (Fig. S10A). Furthermore, gene expression of *adipose triglyceride lipase* (*Atgl*), a pivotal regulator for lipolysis (Duncan RE *et al*, *Annu Rev Nutr* 27:79-101,

2007), showed similar changed pattern (Fig. S10B). These results suggest that knocking down of GCN2 in amygdala prevented leucine deprivation-stimulated lipolysis, which may contribute to the prevention of fat loss in these mice.

This information has been added to Results (page 7), Discussion (page 16) and Supplementary Figure (S6) in the revised manuscript.

Supplementary Figure 11. The effect of GCN2 knockdown in amygdala on lipolysis-related genes and proteins in subcutaneous white adipose tissue (sWAT) under leucine deprivation.

A: P-HSL, t-HSL and p-PKA substrates proteins in sWAT by western blotting (left) and quantified by densitometric analysis (right);

B: Gene expression of *Atgl* in sWAT by RT-PCR.

All studies were conducted using 20- to 22-week-old male control mice (GCN2^{+/+}) or mice with GCN2 deletion in amygdala (GCN2 KO) fed a control (Con) or leucine-deficient [(-) L] diet for 3 days. Data are expressed as the mean ± SEM (n = 6–7 mice/group, as indicated), with individual data points. Data were analyzed by one-way ANOVA followed by the SNK (Student–Newman–Keuls) test. **P* < 0.05 for the effect of any group versus control mice under a Con diet; #*P* < 0.05 for the effect of GCN2 KO mice versus control mice both under a (-) L diet.

-Figure S6: Additional controls should be given: Groups without shGCN2 (control group) and animals fed on a control diet should be given (to measure the possible effects of shGCN2 and/or ATF4 overexpression independently on the -leu diet).

Our response:

We agree with the reviewer that additional controls should be included. In this study, we have had groups with GCN2 deletion or ATF4 over-expression in PKC-δ neurons. We found that deletion of GCN2 in PKC-δ neurons had no significant effect on WAT browning and over-expression of ATF4 in PKC-δ neurons promoted WAT browning, when mice were maintained on a control diet, as shown in Figures 3, S3, and S5 in the original manuscript. However, the effect of ATF4 over-expression in

PKC- δ neurons under leucine deprivation has not been tested.

In response to the reviewer's inquiry, we repeated the experiments in Fig. S6 with more controls including groups without shGCN2 or ATF4 under leucine deprivation, as well as mice maintained on a control diet. We injected Cre-dependent AAVs expressing shGCN2 (AAV-Flex-shGCN2-GFP) and ATF4 (AAV-DIO-ATF4-mCherry), as well as control AAVs, as indicated, to CeA of PKC- δ -Cre mice, and fed these mice a control or leucine-deficient diet. Leucine deprivation induced WAT browning, as demonstrated by the corresponding changes in fat mass weight, H&E staining, as well as the expression of markers for WAT browning of sWAT (Fig.S12). Leucine deprivation-induced these changes were blocked by knockdown of GCN2 in PKC- δ neurons, and the blocking effect of GCN2 knockdown were then reversed by over-expression of ATF4 in these neurons (Fig.S12). As observed in mice under a control diet, over-expression of ATF4 in PKC- δ neurons also promoted WAT browning (Fig. S12).

These results suggest that ATF4 functions as a downstream signal for GCN2 in PKC- δ neurons to regulate WAT browning under leucine deprivation. We also felt that including more controls would give better picture for understanding the role of GCN2/ATF4 pathway in this regulation. Therefore, we have replaced the original Fig. S6 with new data in Supplementary Figure (S14) and modified the relevant descriptions in Results (page 11) in the revised manuscript.

Supplementary Figure 12. Metabolic parameters related to mice with GCN2 knockdown with or without ATF4 over-expression in amygdalar PKC- δ neurons under leucine deprivation.

A: Fat mass by NMR;

B: Adipose tissue weight;

C: Representative images of hematoxylin and eosin (H&E) staining of subcutaneous white adipose tissue (sWAT) and epididymal WAT (eWAT);

D: Gene expression of *Ucp1*, *Pgc1 α* , *Cidea*, *Dio2* and *Prdm16* in sWAT by RT-PCR;

E: UCP1 protein in sWAT by western blotting (left) and quantified by densitometric analysis (right);

F: Representative images of immunohistochemistry (IHC) of UCP1 in sWAT.

Studies were conducted using 20- to 22-week-old male PKC- δ -Cre mice receiving AAVs expressing GFP and mCherry (PKC δ - shGCN2 - ATF4), shGCN2 and mCherry (PKC δ + shGCN2 - ATF4), GFP and ATF4 (PKC δ - shGCN2 + ATF4), or shGCN2 and ATF4 (PKC δ + shGCN2 + ATF4), fed a leucine-deficient [(-) L] diet for 3 days; or receiving AAVs expressing GFP and mCherry (PKC δ - shGCN2 - ATF4) fed a control diet (Con) for 3 days. Data are expressed as the mean \pm SEM (n = 6-7 mice/group as indicated), with individual data points. Data were analyzed by one-way ANOVA followed by the SNK (Student–Newman–Keuls) test. *P < 0.05 for the effect of any group versus PKC δ - shGCN2 - ATF4 mice under a Con diet; #P < 0.05 for the effect of PKC δ + shGCN2- ATF4 mice versus PKC δ - shGCN2 - ATF4 mice both under a (-) L diet; &P < 0.05 for the effect of PKC δ + shGCN2 + ATF4 versus PKC δ + shGCN2 - ATF4 both under a (-) L diet.

-In discussion It is written: “In the hypothalamus, ATF4 probably is an important ER stress regulator; while in the amygdala, ATF4 mostly likely functions as an amino acid sensor. » Could you please give more details.

Our response:

We appreciate it very much for the reviewer's pointing out this issue. Because decreased ER stress in hypothalamus is shown to be associated with or activates WAT browning (Martínez-Sánchez N et al, *Cell Metab* 26(1):212-229, 2017; Contreras C et al, *Diabetes* 66(1):87-99, 2017) and ATF4 is a well-known regulator for ER stress (Balsa E et al, *Mol Cell* 74(5):877-890, 2017), suggesting that ATF4 is likely to function as a ER stress regulator to control energy homeostasis in the hypothalamus. This possibility remains to be investigated.

However, we did not observe significant changes in genes and proteins related to ER stress in amygdala under leucine deprivation (see our response to above questions from the second reviewer), suggesting that amygdalar ATF4 is unlikely to function via ER stress in this case. Because ATF4 functions as downstream signal for amino acid sensor GCN2 in amygdala to regulate WAT browning under leucine deprivation, we proposed that amygdalar ATF4 is likely to regulate WAT browning as an amino acid-responsive gene under leucine deprivation.

We are sorry for the confusion and have added more details regarding this issue to Discussion (pages 15 and 16) in the revised manuscript.

Reviewer #3 (Remarks to the Author):

In this study, the authors identify an interesting new role of the PKC δ neurons of the central amygdala, and identify intracellular effectors including amino sensor GCN2 and transcription factor ATF4. While very interesting and performed thoroughly, the presentation of the results is not well organized, clarity could be improved, and the statistical analysis needs to be revised. Also, the manuscript is filled with English grammatical errors (I listed a sample in the minor comments). The text needs to be proofread by a native English speaker and/or a scientific writer.

First, as the main finding of the study is the role of the role of PKC δ in WAT browning under Leucine deprivation, it would be relevant to present the experiments reporting the role of this neural populations in this metabolic response, prior to describing the experiments testing the role of intracellular effectors (GCN2 and ATF4).

Our response:

We appreciate it very much for the reviewer's pointing out this issue. We also felt that presenting data regarding the role of PKC- δ neurons prior to testing the effect

of GCN2 and ATF4 would make the flow of our manuscript sound better. Therefore, we have rearranged the order of data presenting as suggested.

Second, the statistical analysis is not appropriate. The author use multiple Student t-tests without correcting for multiple comparisons, and should consider performing ANOVA rather than t-tests. Overall, the individual data points on every histogram should also be represented to represent the individual variability of their measures.

Our response:

We appreciate it very much for the reviewer's pointing out this issue. We agree with the reviewer that we should perform ANOVA rather than t-tests. As suggested, the differences for multiple comparisons were assessed by one way ANOVA followed by the SNK (Student–Newman–Keuls) test. In addition, the individual data points on every histogram were also shown to represent the individual variability of their measures. Statistical analyses were performed in GraphPad Prism. This information has been added to Methods (page 24) and all of the Figures where it concerns.

Finally, a summary diagram/schematic would be extremely valuable for the accessibility of the manuscript. (Leucine deprivation  PKCd neurons in CeA[GCN2  eIF2a  ATF4]  ...  SNS  WAT browning)

Our response:

We appreciate it very much for the reviewer's suggestion and have drawn a summary diagram (see our response to question No. 2 from the second reviewer). This information has been added to Discussion (page 17) and Figures (7K).

Minor Comments_____

As Leucine biochemical symbol is L, the author could use this abbreviating in the manuscript: (—)L rather than (—)Leu.

Our response:

As suggested, changes have been made in all of the Figures (including Supplementary Figures) and Figure Legends (including Supplementary Figure Legends).

Multiple abbreviations are not defined: IF (immunofluorescence), H&E (Haemotoxylin and Eosin). Similarly, GCN2 and ATF4 are used in the abstract and not defined there. The role of ATF4 is even not described.

Our response:

In response to the reviewer's inquiry, we have defined the abbreviations for those missing. As suggested, we have also added a sentence describing the role of ATF4 as "an amino acid response gene" in Abstract (page 1).

Figure 5A: Please add the unit to the graph (%)

Our response:

We appreciate it very much for the reviewer's pointing out the unit of Figure 5A in the original figures was not so clear. Actually, the unit of the statistics indicated the number of cells in which c-Fos/tdTomato were co-localized (Fig. S13). To make it more clearly, we have modified the unit in Figures and the relevant description in Figure Legends (page 30) in the revised manuscript. Similar corrections have also been made to Figures (2A and 4A) and Supplementary Figures (S3B and S12A) in the revised manuscript.

Supplementary Figure 13. c-Fos staining in the amygdalar PKC- δ neurons during leucine deprivation.

Immunofluorescence (IF) staining for tdTomato (red), c-Fos (green) and merge (yellow) in central amygdala (CeA) sections (left), and quantification of c-Fos and tdTomato colocalized cell numbers (right).

Studies were conducted using 12- to 14-week-old male PKC- δ -Cre/Ai9 mice fed a control (Con) or leucine-deficient [(-) L] diet for 3 days. Data are expressed as the mean \pm SEM ($n = 6$ mice/group), with individual data points. Data were analyzed by two-tailed unpaired Student's t test. $*P < 0.05$ for the effect of (-) L group versus Con group.

Figure 7: Please correct the legend. The grey and black squares on top of the figure seem to be swapped.

Our response:

Change has been made as suggested.

Figure 7H: Please correct the y axis numbers.

Our response:

Change has been made as suggested.

Line 11-13: Please rephrase, the sentence is hardly understandable

Our response:

We appreciate it very much for the reviewer's pointing out this issue. In response to the reviewer's inquiry, we have rewritten the sentence as the following: Here, we showed that leucine deficiency induced WAT browning, largely blocked by PKC- δ neuronal activity inhibition and adeno-associated virus (AAV)-mediated amygdalar GCN2 deletion. Furthermore, knockdown of GCN2 in amygdalar PKC- δ neurons blocked leucine deprivation-induced WAT browning, reversed by AAV-mediated amino acid responsive gene activating transcription factor 4 (ATF4) over-expression and mediated by altering the activities of amygdalar PKC- δ neurons and sympathetic nervous system.

We are sorry for the confusion and have modified the description in Abstract (page 1).

Line 16: "...of PKC-d neuron and..." Please specify "...of PKC-d neuron in the central amygdala and..."

Our response:

Change has been made as suggested.

Line 44 "on serine 51" is unnecessary, please remove

Our response:

Change has been made as suggested.

Line 45: please detail the effectors of the adaptive responses.

Our response:

We appreciate it very much for the reviewer's pointing out this issue. In response to the reviewer's inquiry, we have rewritten the sentence as the following: General control nonderepressible 2 (GCN2) is an ancient protein kinase that senses intracellular amino acid deficiencies, which then couples the accumulation of uncharged transfer RNAs (tRNAs) to the phosphorylation of eukaryotic initiation factor 2 α (eIF2 α) and thereby increases translation of mRNAs for several effectors that exert many functions, such as regulating amino acid biosynthesis and transports, as adaptive responses (Gietzen, D. W. et al, *Molecular neurobiology* 46, 332-348, 2012). One of such examples is the amino acid response gene activating transcription factor 4 (ATF4) (Gietzen, D. W. et al, *Molecular neurobiology* 46, 332-348, 2012).

We are sorry for the confusion and have modified the description in Introduction (page 3).

Line 47: regulating the other • regulating other

Our response:

We appreciate it very much for the reviewer's suggestion. This sentence has been modified.

Line 48: delete “if there is the possibility”

Our response:

Change has been made as suggested.

Line 50: replace ways with a more scientific term, such as effectors

Our response:

Change has been made as suggested.

Line 51: please cite some external stimuli

Our response:

We appreciate it very much for the reviewer's pointing out this issue. In response to the reviewer's inquiry, we have rewritten the sentence as the following: Activation of the sympathetic nervous system (SNS) is one of the major ways mediating the effects of various external stimulus, such as cold exposure and exercise (Wang, B. et al. *EMBO reports* 19, 2018; Aldiss, P et al, *Metabolism: clinical and experimental* 81,63-70, 2018), on WAT browning.

We are sorry for the confusion and have modified the description in Introduction (page 2).

Line 56: three parts = three main parts. Indeed the amygdala also includes the basomedial and cortical amygdala.

Our response:

Change has been made as suggested.

Line 58: please specify some of the cognitive function of the amygdala.

Our response:

We appreciate it very much for the reviewer's pointing out this issue. In response to the reviewer's inquiry, we have added some of the cognitive function of the amygdala, such as memory and social behavior (Janak PH et al, *Nature* 517(7534):284-92, 2015).

We are sorry for the confusion and have modified the description in Introduction (page 3).

Line 60: mediate the • mediate

Our response:

Change has been made as suggested.

Line 63: our current • this

Our response:

Change has been made as suggested.

Line 153 “” Please be cleared. Do the author refer to the pathway ? Do these two proteins form a complex ?

Our response:

What we mean is the GCN2/eIF2 α "pathway" and we have corrected the description.

Line 155: the sentence is grammatically wrong. Please replace the comma with a period. (“. We speculate...”)

Our response:

Change has been made as suggested.

Line 156: as downstream what ? effector ?

Our response:

Yes, as downstream effector as mentioned by the reviewer. We have corrected the description as suggested.

Lin 162, 173, 214: please replace “asked” with “tested” . The authors did not only asked, they performed experiments the experiments to test the hypotheses.

Our response:

Change has been made as suggested.

Line 197: do the authors mean alteration rather than alternation ?

Our response:

Yes, we mean "alteration" as mentioned by the reviewer. We have corrected the description as suggested.

Line 203: The authors inappropriately mention they pharmacologically inhibit, when they chemogenetically inhibit. Please correct accordingly.

Our response:

Change has been made as suggested.

Line 215: please correct “activated the activity” with “increased the activity” ...

Our response:

Change has been made as suggested.

Line 226: “in THE brain”

Our response:

Change has been made as suggested.

Line 234: pharmacologic = pharmacotoxic

Our response:

Change has been made as suggested.

Line 273: amino acid = amino acids

Our response:

Change has been made as suggested.

Line 295: mostly likely = most likely

Our response:

Change has been made as suggested.

Reviewer #4 (Remarks to the Author):

In this manuscript, the authors focused on the metabolic effects of leucine deprived diet (LDD) and revealed an interesting and important neural mechanism of LDD -induced white adipose browning. By combining multiple viral and mouse genetic tools and chemogenetic DREADDs, the authors demonstrated that LDD induced white adipose browning and adipose loss in mice. They further observed that GCN2 activation in the amygdala PKC- δ neurons and GCN2-engaged ATF4 signaling are both necessary and required to mediate LDD' s such effects. Overall, the results and methods in the current manuscript are novel and findings are significant. My comments are listed below:

1. According to prior studies, LDD negatively regulates body weight by reducing food intake and increasing BAT energy expenditure. Here the authors showed further that WAT-browning is also involved. How much the effect on body weight loss from LDD is actually mediated by WAT-browning,

compared with that from feeding inhibition and BAT activation? Is browning a significant component for LDD to regulate BW? The authors showed data of reduced fat mass but whether such fat mass loss is indeed mediated by WAT-browning is not clear. In addition, CeA neurons and the PKC- δ neurons in the brain region were both shown to regulate feeding and cause anorexia. When the authors manipulated these neurons by deleting/knocking down GCN2, DN- or overexpression of ATF4, and by excitatory and inhibitory DREADDs, did feeding behaviors also change? If yes, how the authors would distinguish the physiologic effects from CeA-mediated anorexia vs. browning? If no, how would the authors explain that LDD-induced fat mass loss was completely blocked in CeA-GCN2 KO/KD, DN-ATF4, and in hM4Di related studies?

Our response:

It is shown that leucine deprivation decreases body weight, associated with decreased food intake and increased energy expenditure, and the increased energy expenditure is likely to be caused by increased thermogenesis in BAT and lipolysis in WAT (Cheng Y *et al*, *Diabetes* 59(1):17-25, 2010). In our current study, we show that WAT browning is also induced by leucine deprivation. All of these four components (but not limited to them) may contribute to the fat loss under leucine deprivation. By conducting pair-feeding experiments, it has been previously shown that food intake is unlikely to play a major role in leucine deprivation-decreased body weight and fat mass (Cheng Y *et al*, *Diabetes* 59(1):17-25, 2010). The relative contribution of BAT and WAT to fat mass and body weight under leucine deprivation has not been tested.

In our current study, we found that fat mass reduction was prevented in mice with GCN2 deletion in amygdala under leucine deprivation. Then we investigated the possible mechanisms underlying this blocking effect. Because CeA neurons and the PKC- δ neurons in the brain region are both shown to regulate feeding (Cai H *et al*, *Nature neuroscience* 17, 1240-1248, 2014; Isosaka, T. *et al*. *Cell* 163, 1153-1164, 2015), food intake might change following different treatments. As predicted, food intake reduction by leucine deprivation was partly blocked in mice with GCN2 deletion in amygdala (as well as in other mice including PKC- δ -shGCN2, PKC- δ -DN-ATF4 and PKC- δ -hM4Di mice) (Fig. S14), which might contribute to the fat retention effect in these mice. To evaluate the contribution of the prevented reduction in food intake to fat loss and body weight, we performed a pair-feeding experiment in PKC- δ -hM4Di mice by providing them with the same amount of food intake as observed in mice without injection of hM4Di under leucine deprivation. The fat mass and body weight did not change significantly in pair-fed PKC- δ -hM4Di mice compared with freely fed PKC- δ -hM4Di mice (Fig. S15). These results suggest that the less reduced food intake under each different treatment is unlikely to contribute to the fat retention effect under leucine deprivation.

Furthermore, we found that UCP1 expression was comparably induced by leucine deprivation in BAT of both control mice and mice with GCN2 deletion in amygdala (see our response to question No. 1 from the first reviewer). However, this

fat retention effect was associated with inhibition of WAT browning, suggesting that WAT browning may be involved in leucine deprivation-reduced fat mass and body weight. We have also measured indirect calorimetry using a comprehensive lab animal monitoring system and found that leucine deprivation-increased oxygen consumption and energy expenditure was also partly reduced by deletion of GCN2 in amygdala (see our response to question No. 1 from the first reviewer). These results suggest that WAT browning may play a role in leucine deprivation-reduced fat mass and body weight. However, what makes it complicated is lipolysis (Cheng Y *et al*, *Diabetes* 59(1):17-25, 2010) and browning in WAT are both controlled by the sympathetic nervous system under leucine deprivation. When WAT browning was blocked by deletion of GCN2 in amygdala, lipolysis in WAT was also blocked (see our response to question regarding lipolysis from the second reviewer) under leucine deprivation. The blocked lipolysis in WAT may also contribute to the fat retention effect in Amy-GCN2 KO mice, which is difficult to distinguish in our study and need to be studied in the future.

Taken together, these results suggest that WAT browning may play a role in leucine deprivation-reduced fat mass and body weight, though our current study could not quantitatively distinguish the contribution of WAT browning and other factors to leucine deprivation-induced effect. The relative contribution of WAT browning to the fat loss under leucine deprivation will be studied in the future.

This information has been added to Discussion (pages 16 and 17) and Supplementary Figure (S18) in the revised manuscript.

Supplementary Figure 14. Leucine deprivation-reduced food intake is partly blocked in several types of mice.

A-D: Daily food intake.

Studies for A were conducted using 20- to 22-week-old male control mice (GCN2^{+/+}) or mice with GCN2 deletion in amygdala (GCN2 KO) fed a control (Con) or leucine-deficient [(-) L] diet for 3 days; studies for B were conducted using 13- to

16-week-old male PKC- δ -Cre mice receiving AAVs expressing GFP (PKC δ - shGCN2) or shGCN2 (PKC δ + shGCN2) fed a Con or (-) L diet for 3 days; studies for C were conducted using 13- to 16-week-old male PKC- δ -Cre mice receiving AAVs expressing mCherry (PKC δ - DN ATF4) or DN ATF4 (PKC δ + DN ATF4) fed a Con or (-) L diet for 3 days; studies for D were conducted using 22- to 24-week-old male PKC- δ -Cre mice receiving AAVs expressing mCherry (PKC δ - hM4Di) or hM4Di (PKC δ + hM4Di), all received CNO injections every 12h for 3 days, fed a Con or (-) L diet for 3 days. Data are expressed as the mean \pm SEM (n = 6-7 mice/group as indicated), with individual data points. Data were analyzed by one-way ANOVA followed by the SNK (Student–Newman–Keuls) test. * P < 0.05 for the effect of any group versus control mice under a Con diet; # P < 0.05 for the effect of GCN2 KO mice, PKC δ + shGCN2 mice, PKC δ + DN ATF4 mice, or PKC δ + hM4Di mice versus control mice all under a (-) L diet.

Supplementary Figure 15. The effect of pair-feeding to the blocking effect on fat loss in PKC- δ -hM4Di mice under leucine deprivation.

A: Daily food intake;

B: Body weight change relative to original body weight;

C: Fat mass by NMR;

D: Subcutaneous WAT (sWAT) and epididymal with adipose tissue (eWAT) weight.

Studies were conducted using 15- to 16-week-old male PKC- δ -Cre mice receiving AAVs expressing mCherry (PKC δ - hM4Di) or hM4Di (PKC δ + hM4Di), fed a control (Con) or leucine-deficient [(-) L] diet for 3 days' pair-feeding. Data are expressed as the mean \pm SEM (n = 6-7 mice/group as indicated), with individual data points. Data were analyzed by one-way ANOVA followed by the SNK (Student–Newman–Keuls) test. * P < 0.05 for the effect of any group versus control mice under a Con diet; # P < 0.05 for the effect of PKC δ + hM4Di mice with or without pair-fed versus control mice both under a (-) L diet.

2. Fig S1G/S2D: LDD induced cFos in both CeA and BLA and it seems that BLA had more notable c-Fos expression, together with stronger expression

of GCN2. How BLA is involved? Since antibodies for GCN2/ATF4 worked well for IF, how many c-Fos neurons are actually GCN2+?

Our response:

The main discovery of our study is that GCN2 in CeA PKC- δ neurons regulates WAT browning under leucine deprivation via affecting neuronal activity. However, as mentioned by the reviewer, leucine deprivation also increased c-Fos expression in BLA (Fig. S1G in the original manuscript), where with quite high expression of GCN2 (Fig. S2D), suggesting that GCN2 in BLA may also be involved in the regulation of WAT browning under leucine deprivation. We appreciate it very much for the reviewer's bringing up this interesting issue.

GCN2 is generally expressed in most of the cells and tissues, however, it becomes activated after being phosphorylated (Gietzen, D. W. et al, *Molecular neurobiology* 46, 332-348, 2012). To investigate a role of GCN2 in this regulation, it will be necessary to examine the activation of GCN2 first. In response to the reviewer's inquiry, we conducted IF staining to see whether GCN2 in BLA was activated by leucine deprivation in mice maintained on a control or leucine-deficient diet. Increased staining of p-GCN2, as well as c-Fos, were observed in BLA of mice maintained on a leucine-deficient diet compared with control mice, however, most of the increased p-GCN2 and c-Fos were not overlapped (Fig. S16A-D). The increased p-GCN2 suggested that GCN2 in BLA may be involved in the regulation of WAT browning under leucine deprivation. However, because only small amount of activated GCN2 was overlapped with c-Fos, suggesting that GCN2 in BLA was either unlikely to play an important role in WAT browning or regulate WAT browning via a neuronal activity-independent pathway under leucine deprivation. These possibilities, however, require systemic and comprehensive analysis to be demonstrated, which will be studied in the future.

This information has been added to Discussion (page 14) in the revised manuscript.

Supplementary Figure 16. Immunofluorescence (IF) staining of c-Fos and p-GCN2 in the basolateral nuclei of the amygdala (BLA) of mice under leucine deprivation

A: IF staining for c-Fos (red), p-GCN2 (green) and merge (yellow) in BLA;

B: Quantification of c-Fos neurons in BLA;

C: Quantification of p-GCN2 neurons in BLA;

D: Quantification of the percentage of p-GCN2 positive (p-GCN2+/c-Fos) or negative (p-GCN2-/c-Fos) neurons in c-Fos expressing neurons in BLA under leucine deprivation.

Studies were conducted using 14- to 15-week-old male wild-type mice fed a control (Con) or leucine-deficient [(-) L] diet for 3 days. Data are expressed as the mean \pm SEM (n = 6 mice/group), with individual data points. Data were analyzed by two-tailed unpaired Student's t test. * $P < 0.05$ for the effect of (-) L versus Con diet group.

3. Fig S1E: Dramatic variations of Ucp1 expression were observed in the upper panel western blot results, which are obviously NOT consistent with the quantified bar data provided below.

Our response:

We appreciate it very much for the reviewer's careful reading of our manuscript. We agree that the variation of UCP1 expression was quite dramatic within the same treatment group. In response to the reviewer's inquiry, we repeated this experiments with more samples (original n = 6, now n= 12) and found that UCP1 protein expression was increased in eWAT of leucine-deprived mice, which were consistent with changes as shown for quantification (Fig. S17). We are sorry for the confusion and have replaced the original Fig. S1E with the new data. This information has also been added to Supplementary Figure Legends (page 1).

Supplementary Figure 17. UCP1 expression in epididymal white adipose tissue (eWAT) of mice under leucine deprivation.

Figures represent UCP1 protein in eWAT by western blotting (left) and quantified by densitometric analysis (right).

Studies were conducted using 14- to 15-week-old male wild-type mice fed a control (Con) or leucine-deficient [(-) L] diet for 3 days. Data are expressed as the mean \pm SEM (n = 12 mice/group), with individual data points. Data were analyzed by two-tailed unpaired Student's t test. * $P < 0.05$ for the effect of (-) L versus Con diet group.

4. Line 112-113: this sentence is confusing. I think the authors were trying to say that AAV-injected GCN2+/+ mice used as control and AAV-injected GCN2lox/lox mice (i.e. GCN2 KO mice) as the study objects. Or, AAV-GFP virus injected GCN2lox/lox mice were used as control. No matter what, a clearer description is required.

Our response:

We appreciate it very much for the reviewer's careful reading of our manuscript. What we mean is that AAV-CAG-GFP injected GCN2^{loxp/loxp} (GCN2^{+/+}) mice were used as control and AAV-CAG-Cre-GFP injected GCN2^{+/+} mice (i.e. GCN2 KO mice) were used as the study objects.

In response, we have modified the description to "We stereotaxically injected adeno-associated virus (AAV)s expressing Cre-GFP or GFP into the amygdala of GCN2 loxp/loxp (GCN2^{+/+}) mice (Fig. S5A) to delete GCN2 only in the amygdala (GCN2 KO) or act as control". In addition, we have modified the relevant description in Methods: GCN2 loxp/loxp (GCN2^{+/+}) mice were bilaterally injected into the amygdala with an AAV vector containing a cassette expressing Cre recombinase protein with GFP (AAV9-CAG-Cre-GFP; 3.5×10^{12} Pfu/mL) at a volume of 250 nl for each side, or a AAV vector containing a cassette expressing GFP protein (AAV9-CAG- GFP; 3.5×10^{12} Pfu/mL) at the same volume as control.

5. Fig S2B: The authors used the gene changes in the thalamus as control. However, a much better control shall be the arcuate hypothalamus, which contains important neurons expressing both GCN2 and ATF4 and previously shown to mediate both LDD' s effects and WAT-browning.

Our response:

We appreciate it very much for the reviewer's good suggestion. In response to the reviewer's inquiry, we examined the expression of *Gcn2* via RT-PCR in the arcuate nucleus (ARC) of the hypothalamus of mice maintained on a control or leucine-deficient diet. As observed for thalamus, no difference in *Gcn2* expression was observed in ARC of mice under leucine deprivation (Fig. S18A). Consistently, GCN2 protein levels were also not changed in ARC of leucine-deprived mice (Fig. S18B).

As suggested, we have removed thalamus data in the original Fig. S2B and S2C, and replaced them with new ARC data (Fig. S5B and S5C in the revised manuscript). We have also modified relevant description in Results (page 7) in the revised manuscript.

Supplementary Figure 18. GCN2 expression in the arcuate nucleus (ARC) of the hypothalamus of mice under leucine deprivation.

A: Gene expression of *Gcn2* in ARC by RT-PCR;

B: GCN2 protein in ARC by western blotting (top) and quantified by densitometric analysis (bottom).

All studies were conducted using 20- to 22-week-old male control mice (GCN2^{+/+}) or mice with GCN2 deletion in amygdala (GCN2 KO). Data are expressed as the mean ± SEM (n = 6 mice/group), with individual data points. Data were analyzed by two-tailed unpaired Student's t test. **P* < 0.05 for the effect of GCN2 KO mice versus control mice.

6. Fig S7B, 5B–H: How CNO + LDD treatment was performed was not clear. How CNO was injected, whether saline injection of the same mice was included as control were unclear, either. More controls groups are required, particularly given the recent findings that CNO has extensive DREADDs-independent effects in the brain.

Our response:

We appreciate it very much for the reviewer's careful reading of our manuscript. As mentioned by the reviewer, it is reported that CNO actually has other DREADDs-independent effects in the brain (Gomez JL et al, *Science* 357(6350):503-507, 2017). For example, CNO can competitively inhibit the binding of ligands at several receptors, including those for histamine H1, 5-HT2A, muscarinic M1 and others (Gomez JL et al, *Science* 357(6350):503-507, 2017).

To avoid the additional effects of CNO to our mice, we carried out the experiments as the following. PKC-δ-Cre mice were bilaterally injected into CeA with 200 nl Cre-dependent AAV encoding only mCherry (AAV-EF1a-DIO-mCherry) as control group, or Cre-dependent AAV encoding hM4Di and mCherry (AAV-EF1a-DIO-hM4Di-mCherry) as experimental group. Four weeks after AAV delivery, both control and experimental groups of mice received intraperitoneal (i.p.) injections with CNO at 5 mg/kg of body weight every 12 h for 3 days as shown previously (Cai, H. et al, *Nature neuroscience* 17, 1240-1248, 2014). Because both control and experimental groups of mice were given CNO, the potential additional effects of CNO should be able to be avoided.

We are sorry for not including enough information in the original manuscript and have added more details to Results (pages 5 and 9) and Methods (pages 20 and 21) in the revised manuscript.

7. Many AAVs were used in the current study. However, serotypes, titer, and construction of AAVs were all missing.

Our response:

We appreciate it very much for the reviewer's good suggestion. All the AAV were produced at the OBiO Technology (Shanghai) in our study.

For GCN2 ablation study, GCN2loxp/loxp (GCN2^{+/+}) mice were bilaterally injected in the amygdala with an AAV vector containing a cassette expressing Cre

recombinase protein with GFP (AAV9-CAG-Cre-GFP; 3.5×10^{12} Pfu/mL), or a AAV vector containing a cassette expressing GFP protein (AAV9-CAG- GFP; 3.5×10^{12} Pfu/mL) as control.

For knocking down GCN2 in amygdalar PKC- δ neurons, PKC- δ -Cre mice were bilaterally injected with a Cre-dependent AAV vector containing the mir-30-shGCN2 coding sequence and GFP protein in the opposite orientation flanked by two inverted loxP sites (AAV9-CMV-bGiobin-FLEX -mir-30-shGCN2-GFP; 4.9×10^{12} Pfu/mL), or a AAV vector containing the mir-30-scramble and GFP protein in the opposite orientation flanked by two inverted loxP sites (AAV9-CMV-bGiobin-FLEX -mir-30-scramble-GFP; dilute to 4.9×10^{12} Pfu/mL) as control. The target sequence is 5'-TCTGGATGGATTAGCTTATA-3' for GCN2.

For inhibiting ATF4 in amygdalar PKC- δ neurons, PKC- δ -Cre mice were bilaterally injected with a Cre-dependent AAV vector containing the dominant-negative form of ATF4 (DN-ATF4), with mutation of 6 amino acids (²⁹²R²⁹¹YRQK²⁹²K²⁹³R²⁹⁴ to ²⁹²G²⁹¹YLEAAA²⁹²) within the DNA-binding domain (HE CH et al, *J Biol Chem* 276(24):20858-65, 2001), in the opposite orientation flanked by two inverted loxP sites (AAV9-EF1a-DIO-DN-ATF4-mCherry, 1.3×10^{12} Pfu/mL), or a AAV vector containing only mCherry in the opposite orientation flanked by two inverted loxP sites (AAV9-EF1a-DIO- mCherry, 1.3×10^{12} Pfu/mL) as control.

For over-expressing ATF4 in PKC- δ neurons, PKC- δ -Cre mice were bilaterally injected with a Cre-dependent AAV vector containing ATF4 in the opposite orientation flanked by two inverted loxP sites (AAV9-EF1a-DIO-ATF4-mCherry, 2.5×10^{12} Pfu/mL), or a AAV vector containing only mCherry in the opposite orientation flanked by two inverted loxP sites (AAV9-EF1a-DIO- mCherry, 2.5×10^{12} Pfu/mL) as control.

For inhibiting the neuronal activity of the amygdalar PKC- δ neurons with DREADDs, PKC- δ -Cre mice were stereotaxically injected with a Cre-dependent AAV encoding an inhibitory DREADD GPCR (hM4Di) (AAV9-EF1a-DIO-hM4Di-mCherry, 3.1×10^{12} Pfu/mL), or a Cre-dependent AAV encoding only mCherry(AAV9-EF1a-DIO- mCherry, 3.1×10^{12} Pfu/mL) as control.

For activating neuronal activity of the amygdalar PKC- δ neurons with DREADDs, PKC- δ -Cre mice were stereotaxically injected with a Cre-dependent AAV encoding an excitatory DREADD GPCR (hM3Dq) (AAV9-EF1a-DIO-hM3Dq-mCherry, 2.6×10^{12} Pfu/mL), or a Cre-dependent AAV encoding only mCherry(AAV9-EF1a-DIO- mCherry, 2.6×10^{12} Pfu/mL) as control.

We are sorry for not including enough information in the original manuscript and have added more details to Methods (pages 18-21) in the revised manuscript.

8. Line 360–361: coordinates listed are confusing. Shall CeA be a part of the amygdala? Why two different coordinates were used?

Our response:

We appreciate it very much for the reviewer's careful reading of our manuscript. CeA is a part of the amygdala, but to delete GCN2 more accurately in amygdala or

CeA, we used different coordinates. To deleting GCN2 in the amygdala, the AAV need to infect the neurons in both CeA and BLA, so we chose the site between CeA and BLA that covered more area of amygdala ($\pm 3.10, -1.42, -4.8$). To investigate the signaling in PKC- δ neurons of CeA, the AAV need to be injected into CeA only to reduce the AAV-induced influence in other areas, therefore we chose to inject using a different coordinate ($\pm 3.00, -1.42, -4.8$). Similar strategy has been used in other studies (Ji G et al, *J Neurosci* 37(6):1378-1393, 2017).

We are sorry for not including enough information in the original manuscript and have added the relevant details to Methods (page 19) in the revised manuscript.

Supplementary Figure 19. Schematic of target area of AAV stereotaxical injections. CeA: the central nuclei of the amygdala; BLA: the basolateral nuclei of the amygdala

9. Please improve the writing and make grammatical corrections.

Our response:

We appreciate it very much for the reviewer's good suggestion. In response to the reviewer's inquiry, we have read through all of the manuscript carefully and corrected all the inappropriate words and grammatical problems.

Reviewers' Comments:

Reviewer #1:

Remarks to the Author:

When reviewing a paper, a major task is to try to improve the paper. After such improvements, the paper may come out different, and conclusions may have moved somewhat. This means that there is a risk that although the authors have done everything that initially was suggested, there may be new revisions upcoming. I am generally reluctant to start new issues in the second refereeing round – but after all, this is Nature Communications, and quite high demands can be set.

In this particular case, what does affect the outcome is the appearance in the paper of food intake data. These data make it clear that the mice have become partly anoxic, they eat less than they need, and they therefore start to use their internal stores of fat for survival. This is seen both as a decrease in fat pad weights and as a change in substrate utilization towards lipid combustion. The mobilization of the lipid stores occurs through stimulation of the sympathetic nerves.

Thus, whereas the authors earlier implied a kind of novel direct effect of the amygdala on the beige fat, the scenario that the data now would support would be the following:

A leucine-free diet affects the amygdala and induces anorexia (why the diet has this effect is not direct understandable). The lack of food induces a negative food balance that leads to stimulation of the sympathetic nervous system that releases norepinephrine in different fat depots, including the beige depot. Adrenergic stimulation of certain of the fat cells in these depots leads always (for no known reason) to the induction of the brownish phenotype in these cells. This is a coherent story within known physiological pathways.

One experiment that could be performed, as it is comparatively simply and short, is to mimic the anorexic effect of the leucine-free diet by pair-feeding mice to obtain the same food intake and measure the browning in those mice. This would help the present story – but as also already pointed out by the authors it is indeed already known that fasting promotes browning.

In addition to this possible experiment, the data on food intake – that have a high explanatory value – must be moved from the supplement into the main text and figures for all the experimental setups, and the changed food intake should be the first effect mentioned in all setups. The title should be expanded by something like “Through activation of GCN2/ATF4 in amygdalar PKC-d neurons, leucine deprivation promotes WAT browning due to induced anorexia and subsequent sympathetic lipid mobilization”, and the abstract should be reformulated accordingly. Also the summary figure should be redrawn so that a direct effect of the amygdala on the SNS is not implied but that the effect is mediated via the anorexia induced.

Additionally: the y-axes on fig S8BC should start at 0.

(It has been confusing that the authors present a double nomenclature of figs – thus what is called “Supplementary Figure 1” in Response to Reviewers is actually Fig. S7, etc.)

Reviewer #3:

Remarks to the Author:

The authors addressed most of my comments, and the manuscript holds now up to the scientific standards of publication. However, the writing still needs to be improved and the figures could greatly be improved as well. The manuscript needs to be read by a native English scientist.

----Abstract----

First sentence: "White adipose tissue (WAT) can become browning"

Why is this an interesting fact? What are the physiological consequences of this browning? It is important to add half a sentence explaining this to pique the interest of the readers.

Third sentence: "Here, we showed that leucine deficiency induced WAT browning, largely blocked by PKC- δ neuronal activity inhibition and adeno-associated virus (AAV)-mediated amygdalar GCN2 deletion."

This sentence is grammatically wrong = there is no VERB! I assume the authors meant: Here, we showed that 'leucine deficiency induced WAT browning' *is* largely blocked by PKC- δ neuronal activity inhibition and adeno-associated virus (AAV)-mediated amygdalar GCN2 deletion.

Fourth sentence: same as third: there is no VERB. I assume the authors meant: Furthermore, knockdown of GCN2 in amygdalar PKC- δ neurons blocked leucine deprivation-induced WAT browning *which was* reversed by over-expression of the amino acid responsive gene activating transcription factor 4 (ATF4), and *is* mediated by altering the activities of amygdalar PKC- δ neurons and sympathetic nervous system.

Also, there is redundancy in those 2 sentences. Please avoid redundancies.

----Figures----

The graphs are very small and numbers/legend are hard to read. Please use all the whit/dead space to make each panel bigger. See example of Figure 1 and 5 optimized (enclosed).

Figures1: F and G should be swapped

Minor comments: 0 does not need decimals (replace 0.0 with 0)

Arbitrary unit can be abbreviated A.U.

Reviewer #4:

Remarks to the Author:

Obviously the authors have carefully designed more experiments, included additional results and control data, and significantly improved the manuscript. My major comments have been addressed. I do not have additional comments.

Reviewer #5:

Remarks to the Author:

The authors have been highly responsive to the extensive comments on the original submission, including the production and inclusion of additional data. The data collectively fit well together and provide convincing evidence that the WAT browning induced by leucine deprivation is mediated by PKCdelta neurons in the amygdala. I have only a few minor comments:

1. The manuscript could use some editorial improvements. In particular, I will point out the first sentence of the abstract: "White adipose tissue (WAT) can become browning...", with a similar sentence early in the introduction. It is more appropriate to say "White fat can become brown", but personally I would edit that sentence more extensively.

2. The IHC images are very small and relatively low resolution, even when I zoom in on the screen. This could be an issue with files available to reviewers (or my error), but some larger high-quality images would be helpful. This is particularly relevant for Figure 2, where there seems to be significant background in the cFos channel, and in general it is difficult to make out the structure of the labeled cells in the tomato channel. If higher quality images will not be available in the final

form, then I suggest that larger, higher resolution images be included in supplemental data. Perhaps just representatives of each staining (tomato, cFos, ATF4, etc).

3. Antibodies to intracellular signaling molecules like GCN2 and ATF4 are notoriously poor, and this would be particularly relevant in the IHC work. The only western blot images have a tight focus on the specific band of interest, and thus it is difficult to know if there are additional non-specific bands that might influence the specificity for IHC.

4. The data suggest that CeA neurons are responding, via GCN2, directly to a fall of leucine levels. I wonder if the authors have ever measured leucine concentrations in serum and brain (or amygdala) in their model, to determine if local leucine concentrations indeed fall and thereby stimulate GCN2. If so adding this point into the manuscript would further support the working model.

5. While the primary interest of the manuscript is the evidence that amygdala neurons can influence WAT browning, I think it would be reasonable to at least address whether any of this work is physiologically relevant. In my opinion, it seems highly unlikely that rodents would ever encounter an environment in which leucine is totally absent yet all other nutrients (and amino acids) are readily available. What physiological advantage does the browning of WAT provide to the animal in the leucine deprived state? Is leucine deprivation is triggering an interesting pathological response that would never actually occur in a physiological setting?

Reviewers' comments:

Reviewer #1 (Remarks to the Author):

When reviewing a paper, a major task is to try to improve the paper. After such improvements, the paper may come out different, and conclusions may have moved somewhat. This means that there is a risk that although the authors have done everything that initially was suggested, there may be new revisions upcoming. I am generally reluctant to start new issues in the second refereeing round – but after all, this is Nature Communications, and quite high demands can be set.

In this particular case, what does affect the outcome is the appearance in the paper of food intake data. These data make it clear that the mice have become partly anoxic, they eat less than they need, and they therefore start to use their internal stores of fat for survival. This is seen both as a decrease in fat pad weights and as a change in substrate utilization towards lipid combustion. The mobilization of the lipid stores occurs through stimulation of the sympathetic nerves.

Thus, whereas the authors earlier implied a kind of novel direct effect of the amygdala on the beige fat, the scenario that the data now would support would be the following:

A leucine-free diet affects the amygdala and induces anorexia (why the diet has this effect is not direct understandable). The lack of food induces a negative food balance that leads to stimulation of the sympathetic nervous system that releases norepinephrine in different fat depots, including the beige depot. Adrenergic stimulation of certain of the fat cells in these depots leads always (for no known reason) to the induction of the brownish phenotype in these cells. This is a coherent story within known physiological pathways.

One experiment that could be performed, as it is comparatively simply and short, is to mimic the anorexic effect of the leucine-free diet by pair-feeding mice to obtain the same food intake and measure the browning in those mice. This would help the present story – but as also already pointed out by the authors it is indeed already known that fasting promotes browning.

In addition to this possible experiment, the data on food intake – that have a high explanatory value – must be moved from the supplement into the main text and figures for all the experimental setups, and the changed

food intake should be the first effect mentioned in all setups. The title should be expanded by something like “Through activation of GCN2/ATF4 in amygdalar PKC-d neurons, leucine deprivation promotes WAT browning due to induced anorexia and subsequent sympathetic lipid mobilization “ , and the abstract should be reformulated accordingly. Also the summary figure should be redrawn so that a direct effect of the amygdala on the SNS is not implied but that the effect is mediated via the anorexia induced.

Our response:

1) Regarding the contribution of the reduced food intake to WAT browning under leucine deprivation: it is true that leucine deprivation reduces food intake (Response Fig.1A) with unknown reasons and stimulates sympathetic nervous system (SNS) activity, however, the correlation between these changes is currently unknown. One of the possibilities could be as the following as proposed by the reviewer: the negative food balance may promote the internal fat utilization via stimulating SNS activity. To test this possibility, mice were fed a control, leucine-deficient or pair-fed (about 20 % reduction) control diet for 3 days (Response Fig.1B). As shown in the manuscript, leucine deprivation for 3 days significantly reduced fat mass (as evaluated by NMR) and the weights of subcutaneous WAT (sWAT) and epididymal WAT (eWAT) (Response Fig. 1C and 1D). H&E staining of sWAT and eWAT also showed decreased lipid droplets following leucine deprivation (Response Fig.1E). Moreover, the expression of WAT browning markers including *uncoupling protein-1 (Ucp1)*, *peroxisome proliferator-activated receptor gamma co-activator 1a (Pgc1a)*, *cell death-inducing DFFA-like effector a (Cidea)*, *deiodinase 2 (Dio2)* and *PR domain containing 16 (Prdm16)* were significantly increased in sWAT and eWAT of mice under leucine deprivation compared with mice under a control diet (Response Fig.1F and 1H). Similar change was found in UCP1 proteins (Response Fig.1G and 1I). Different from those observed under leucine deprivation, pair-feeding had no significant effects on all the above parameters examined compared with control mice (Response Fig.1B-I).

Then we investigated whether the reduced food intake would stimulate SNS activity by examining the levels of norepinephrine (NE) in sWAT. NE levels were increased in sWAT of leucine-deprived mice, but not changed in pair-fed mice, compared with mice under a control diet (Response Fig.1J). Similar changes were observed for tyrosine hydroxylase (TH) proteins and *β-adrenergic receptor 3 (Adrb3)* mRNA expression (Response Fig.1K and 1L).

These results suggest that leucine deprivation-induced activation of SNS and induction of WAT browning are unlikely to be caused by the reduced food intake. In support of our results, other studies also show that the reduced food intake decreases SNS activity (Almundarij TI et al, *Physiol Rep* 5 (4), 2017; Müller MJ et al. *Am J Clin Nutr* 102 (4), 807-19 2015).

2) Regarding the effect of fasting on WAT browning: we are sorry to give the reviewer the impression that fasting promotes WAT browning. In fact, different patterns of fasting treatment have different effects on WAT browning. For example,

intermittent fasting (2 days feeding followed by 1 day fasting for 16 weeks) stimulates WAT browning (Kim, K.H. et al, *Cell research* 27, 1309-1326, 2017) as mentioned in our manuscript, whereas continuous fasting for 16 h inhibits UCP1 expression in WAT (Ding H et al, *Nature communications* 7, 11533, 2016). In contrast, our pair-feeding experiments showed that the reduced food intake (about 20 % reduction for 3 days) had no significant effect on WAT browning under leucine deprivation. Therefore, the effect of fasting on WAT browning could be very complicated, need to be evaluated under different conditions.

3) Regarding the data and descriptions of food intake under leucine deprivation: we agree with the reviewer that food intake data are very important to understand the amygdala control of WAT browning under leucine deprivation. As suggested, we have moved food intake data from the supplement into the main figure and mentioned the changed food intake first in all setups, and added food intake-related description in Abstract. In addition, we have added the food intake data the first time mentioned for mice under leucine deprivation to the main Figure 1. We appreciate it very much for the reviewer's suggestions regarding the role of food intake to WAT browning. However, because the main focus of our current study is about amygdala control of WAT browning, the relative contribution of the reduced food intake to the activation of SNS activity and induction of WAT browning was unlikely to be that significant (based on results of pair-feeding experiments), and the molecular mechanisms underlying the reduced food intake is unknown, we prefer not to add food intake-relevant descriptions to Title and Summary Figure in our current study. Despite of these facts, we are quite interested in investigating the molecular mechanisms underlying anorexia induced by leucine deprivation, which will be studied in the future.

This information has been added to Abstract (page 1), Introduction (page 2), Results (pages 4, 6, 7, 9 and 11), Discussion (page 17), Figures (Fig. 1B, 2B, 3B, 4B and 5B), Figure Legends (pages 30, 32, 33 and 37) and Supplementary Figures (Fig. S19).

Response Figure 1. Pair-feeding has no significant impact on WAT browning or sympathetic nervous system.

A: Daily food intake;

B: Daily food intake;
C: Fat mass by NMR;
D: Subcutaneous WAT (sWAT) and epididymal WAT (eWAT) weight;
E: Representative images of hematoxylin and eosin (H&E) staining of sWAT and eWAT;
F: Gene expression of *Ucp1*, *Pgc1a*, *Cidea*, *Dio2* and *Prdm16* in sWAT by RT-PCR;
G: UCP1 protein in sWAT by western blotting (left) and quantified by densitometric analysis (right);
H: Gene expression of *Ucp1*, *Pgc1a*, *Cidea*, *Dio2* and *Prdm16* in eWAT by RT-PCR;
I: UCP1 protein in eWAT by western blotting (left) and quantified by densitometric analysis (right);
J: Norepinephrine (NE) levels in sWAT measured by ELISA kit;
K: TH protein in sWAT by western blotting (left) and quantified by densitometric analysis (right);
L: Gene expression of *Adrb3* in sWAT by RT-PCR.

Studies for A were conducted using 14-week-old male wild-type (WT) mice fed a control (Control) or leucine-deficient [(-) L] diet for 3 days; studies for B-L were conducted using 8-week-old male WT mice fed a Control, (-) L, or pair-fed (Pair-fed) control diet for 3 days. Data are expressed as the mean \pm SEM (n = 5-7 mice/group as indicated), with individual data points. Data were analyzed by two-tailed unpaired Student's t test for A, or by one-way ANOVA followed by the SNK (Student–Newman–Keuls) test for B-L. * $P < 0.05$ for the effect of any group versus mice under a Control diet; # $P < 0.05$ for the effect of a (-) L diet versus Pair-fed diet.

Additionally: the y-axes on fig S8BC should start at 0.

(It has been confusing that the authors present a double nomenclature of figs – thus what is called “Supplementary Figure 1” in Response to Reviewers is actually Fig. S7, etc.)

Our response:

Changes have been made as suggested, which are now Figures S9B and S9C in the revised manuscript. In addition, to avoid any confusion, we have designated the Figures in the response letter as “Response Figure”.

Reviewer #3 (Remarks to the Author):

The authors addressed most of my comments, and the manuscript holds now up to the scientific standards of publication. However, the writing still needs to be improved and the figures could greatly be improved as well. The manuscript needs to be read by a native English scientist.

----Abstract----

First sentence: “White adipose tissue (WAT) can become browning”

Why is this an interesting fact? What are the physiological consequences of this browning? It is important to add half a sentence explaining this to pique the interest of the readers.

Our response:

Thanks for the reviewer's suggestion. In response, we have modified the description as "The browning of white adipose tissue (WAT) has got much attention for its potential beneficial effects on metabolic disorders, however, the nutritional factors and neuronal signals involved remain largely unknown."

Third sentence: "Here, we showed that leucine deficiency induced WAT browning, largely blocked by PKC- neuronal activity inhibition and adeno-associated virus (AAV)-mediated amygdalar GCN2 deletion."

This sentence is grammatically wrong = there is no VERB! I assume the authors meant: Here, we showed that 'leucine deficiency induced WAT browning' *is* largely blocked by PKC- neuronal activity inhibition and adeno-associated virus (AAV)-mediated amygdalar GCN2 deletion.

Our response:

Thanks for the reviewer's suggestion. In response, we have modified the description as the following: "Here, we showed that leucine deficiency could induce WAT browning, which was unlikely to be affected by food intake, but was largely blocked by PKC- δ neuronal activity inhibition and adeno-associated virus (AAV)-mediated amygdalar GCN2 deletion."

Fourth sentence: same as third: there is no VERB. I assume the authors meant: Furthermore, knockdown of GCN2 in amygdalar PKC- neurons blocked leucine deprivation-induced WAT browning *which was* reversed by over-expression of the amino acid responsive gene activating transcription factor 4 (ATF4), and *is* mediated by altering the activities of amygdalar PKC- neurons and sympathetic nervous system.

Our response:

Thanks for the reviewer's careful reading and the changes have been made as suggested.

Also, there is redundancy in those 2 sentences. Please avoid redundancies.

Our response:

Thanks for the reviewer's careful reading. In response, we have modified the description of the Fourth sentence as "Furthermore, GCN2 knockdown in amygdalar PKC- δ neurons blocked WAT browning, which was reversed by over-expression of amino acid responsive gene activating transcription factor 4 (ATF4), and was mediated by the activities of amygdalar PKC- δ neurons and the sympathetic nervous

system". In addition, as suggested, we have asked native English speaker to correct our writing.

----Figures----

The graphs are very small and numbers/legend are hard to read. Please use all the whit/dead space to make each panel bigger. See example of Figure 1 and 5 optimized (enclosed).

Figures1: F and G should be swapped

Minor comments: 0 does not need decimals (replace 0.0 with 0)
Arbitrary unit can be abbreviated A.U.

Our response:

Thanks for the reviewer's careful reading and suggestion. We have made the corrections for all of the Figures as suggested.

Reviewer #4 (Remarks to the Author):

Obviously the authors have carefully designed more experiments, included additional results and control data, and significantly improved the manuscript. My major comments have been addressed. I do not have additional comments.

Reviewer #5 (Remarks to the Author):

The authors have been highly responsive to the extensive comments on the original submission, including the production and inclusion of additional data. The data collectively fit well together and provide convincing evidence that the WAT browning induced by leucine deprivation is mediated by PKCdelta neurons in the amygdala. I have only a few minor comments:

1. The manuscript could use some editorial improvements. In particular, I will point out the first sentence of the abstract: "White adipose tissue (WAT) can become browning...", with a similar sentence early in the introduction. It is more appropriate to say "White fat can become brown", but personally I would edit that sentence more extensively.

Our response:

We appreciate it very much for the reviewer's suggestion and have modified the description as the following: "The browning of white adipose tissue (WAT) has got much attention for its potential beneficial effects on metabolic disorders, however, the nutritional factors and neuronal signals involved remain largely unknown." As

suggested, we have also modified the description of the relevant sentence in Introduction. In addition, we have asked native English speaker to correct our writing.

2. The IHC images are very small and relatively low resolution, even when I zoom in on the screen. This could be an issue with files available to reviewers (or my error), but some larger high-quality images would be helpful. This is particularly relevant for Figure 2, where there seems to be significant background in the cFos channel, and in general it is difficult to make out the structure of the labeled cells in the tomato channel. If higher quality images will not be available in the final form, then I suggest that larger, higher resolution images be included in supplemental data. Perhaps just representatives of each staining (tomato, cFos, ATF4, etc).

Our response:

We agree with the reviewer that larger, higher resolution images should be included in the manuscript. In addition, we agree with the reviewer that the background in the c-Fos channel seemed to be significant. In response, we have repeated c-Fos staining under optimized experimental conditions that have achieved much better quality images (Response Figure 2). As suggested, we have also provided larger, higher resolution images for all of the IHC staining results in the Supplementary Figure (Fig. S20).

Response Figure 2. Immunofluorescence (IF) staining of c-Fos under leucine deprivation with high resolution.

IF staining for tdTomato (red), c-Fos (green) or merge (yellow) in the central amygdala sections. Studies were conducted using 12- to 14-week-old male PKC- δ -Cre/Ai9 mice fed a control (Control) or leucine-deficient [(-) L] diet for 3 days.

3. Antibodies to intracellular signaling molecules like GCN2 and ATF4 are notoriously poor, and this would be particularly relevant in the IHC work. The only western blot images have a tight focus on the specific band of interest, and thus it is difficult to know if there are additional non-specific bands that might influence the specificity for IHC.

Our response:

We appreciate it very much for the reviewer's concern about the specificity of antibodies for GCN2 or ATF4 used in our study. To specifically reflect the changes of GCN2 or ATF4 in IHC staining, we chose antibodies that have designated to be used for IHC. Furthermore, to evaluate the specificity of the antibodies for GCN2 and ATF4 in the IHC work, we have conducted a series of preliminary experiments.

To validate the specificity for GCN2 antibodies, we examined GCN2 expression by IHC in control mice and GCN2 global knockout (KO) mice. IF staining showed that GCN2 was detected in the amygdala of control mice, but absent in GCN2 KO mice (Response Figure 3A). Similar results were obtained in the staining of the whole brain slices (Response Figure 3B).

Because of the lack of ATF4 KO mice, we validated the specificity for ATF4 antibodies in mice following AAV-injection of Cre-dependent ATF4 or control AAV in the amygdalar PKC- δ neurons. As shown in Figure S14D in the revised manuscript (the data was provided here as Response Figure 3C to be convenient for the reviewer to see), IF staining exhibited apparent more fluorescence in the amygdalar PKC- δ neurons overexpressing ATF4 than the control group. The specificity of ATF4 antibodies was then validated using the specific ATF4 blocking peptide (sc-7583P, Santa Cruz Biotechnology, CA, USA). Consistent with our previous results (Figure S13A in the revised manuscript), leucine deprivation increased ATF4 protein levels (as shown by IF staining) in amygdala compared with control mice (Response Figure 3D). But in the presence of ATF4 blocking peptide, no obvious fluorescence was detected in the amygdala of mice either under a control or leucine-deficient diet (Response Figure 3D).

Based on the above results, we believed that the antibodies used were sufficient to reflect the changes of GCN2 and ATF4 in current study.

Response Figure 3. Validation of the specificity for GCN2 and ATF4 antibodies for immunofluorescence (IF) staining.

A: IF staining for GCN2 (green) in amygdala; CeA: the central nuclei of the amygdala; BLA: the basolateral nuclei of the amygdala;

B: Post hoc visualization of GCN2 (green) in the whole brain slices; 3V, third ventricle;

C: IF staining for mCherry (red), ATF4 (green) and merge (yellow) in CeA;

D: IF staining for GCN2 (green) in amygdala.

Studies for A and B were conducted using 12-week-old male wild-type (WT) or GCN2^{-/-} (KO) mice; studies for C were conducted using 13- to 15-week-old male PKC- δ Cre mice receiving AAVs expressing mCherry (PKC δ - ATF4) or ATF4 (PKC δ + ATF4); studies for D were conducted using 14- to 15-week-old male wild-type mice fed a control (Control) or leucine-deficient [(-) L] diet for 3 days.

4. The data suggest that CeA neurons are responding, via GCN2, directly to a fall of leucine levels. I wonder if the authors have ever measured leucine concentrations in serum and brain (or amygdala) in their model, to determine if local leucine concentrations indeed fall and thereby stimulate GCN2. If so adding this point into the manuscript would further support the working model.

Our response:

We appreciate it very much for the reviewer's pointing out this important issue. In fact, we have measured the levels of leucine in the serum and amygdala of mice maintained on a control or leucine-deficient diet for 3 days by high-performance liquid chromatography (Ultimate 3000, USA)-tandem mass spectrometry (API 3200 Q-TRAP, USA). In agreement with the previous reports of 7 days' leucine deprivation (Fei X et al, *Diabetes* 60 (3), 746-56, 2011), leucine deprivation for 3 days also decreased serum leucine levels compared with mice under a control diet (Response

Figure 4A). Similarly, the leucine levels were also reduced in amygdala after leucine deficiency (Response Figure 4B). These results suggest that CeA neurons are responding via GCN2 directly to a fall of leucine levels.

This information has been added to Results (page 6), Working model (Fig. 7K) and Supplementary Figures (Fig. S4).

Response Figure 4. Leucine deprivation decreases leucine levels in serum and amygdala.

Studies were conducted using 14- to 15-week-old male wild-type mice fed a control or leucine-deficient [(-) L] diet for 3 days. Leucine levels were determined by high-performance liquid chromatography (Ultimate 3000, USA)-tandem mass spectrometry. Data are expressed as the mean \pm SEM (n = 3-4 mice/group as indicated), with individual data points. Data were analyzed by two-tailed unpaired Student's t test. * $P < 0.05$ for the effect of a (-) L diet versus control diet group.

5. While the primary interest of the manuscript is the evidence that amygdala neurons can influence WAT browning, I think it would be reasonable to at least address whether any of this work is physiologically relevant. In my opinion, it seems highly unlikely that rodents would ever encounter an environment in which leucine is totally absent yet all other nutrients (and amino acids) are readily available. What physiological advantage does the browning of WAT provide to the animal in the leucine deprived state? Is leucine deprivation is triggering an interesting pathological response that would never actually occur in a physiological setting?

Our response:

1) Regarding the model we used: we agree with the reviewer that in nature it's unlikely that rodents would encounter an environment in which only leucine is totally absent. But we sometimes face the situation when our nutrients are unbalanced, which may result in the lower levels of some amino acids, such as leucine, in the body (Kahleova H et al, *Nutr Diabetes* 8 (1), 58, 2018). Alternatively, a mutation or disorder in the regulation of leucine catabolism enzymes or amino acid transporters may also impact leucine levels (Phillip J W et al, *Cell Metab* 27 (6), 1281-1293.e7, 2018; Wyant GA et al. *Cell* 171 (3), 642-654.e12, 2017). Similar cases were also observed with other amino acids (Amobi A, et al, *Adv Exp Med Biol* 1036, 129-144

2017; Cherqui S and Courtoy PJ, *Nat Rev Nephrol* 13 (2), 115-131, 2017). Even when mice were fed a leucine-deficient diet, it only causes 50 % reduction of leucine levels in the body as shown by others (Fei X et al, *Diabetes* 60 (3), 746-56, 2011) and those of our results. Thus the consequence of “leucine deficiency” may mimic some situations actually occur in nature as mentioned above, therefore our study will help understanding the metabolic networks in response to changes in amino acids.

2) Regarding the physiological or pathological significance: previous studies (Ying Cheng et al, *Diabetes* 59:17-25, 2010) and those of our results (Fig. S8 in the revised manuscript) have shown that leucine deprivation induces BAT UCP1 expression, which stimulates thermogenesis and helps increase the body temperature (Lowell BB et al, *Nature* 404:652-660, 2000; Rosen, E. D. et al, *Cell* 156, 20-44, 2014). Because WAT browning is also considered to increase thermogenesis (Abdullahi et al, *Trends in endocrinology and metabolism* 27, 542-552, 2016; Jiang, H. et al, *Cell metabolism* 26, 686-692 e683, 2017), we speculated that this change might help to increase thermogenesis under leucine deprivation. Therefore, it is possible that leucine deprivation causes a cold-like response that requires thermogenesis to be stimulated in BAT and WAT. Consistent with the possibility, the central thermotaxic center is in the brain, which has been shown to be affected by nutrients including some of the amino acid derivatives (Nakagawa H et al, *J Therm Biol* 58, 15-22, 2016; Murray NM, et al. *Sleep* 38 (12), 1985-93, 2015). These possibilities, however, require to be studied in the future.

We appreciate it very much for the reviewer’s bringing this interesting issue and have added this information to Discussion (page 18).

** See Nature Research’s author and referees’ website at www.nature.com/authors for information about policies, services and author benefits

This email has been sent through the Springer Nature Tracking System NY-610A-NPG&MTSC
Confidentiality Statement: This e-mail is confidential and subject to copyright. Any unauthorised use or disclosure of its contents is prohibited. If you have received this email in error please notify our Manuscript Tracking System Helpdesk team at <http://platformsupport.nature.com>. Details of the confidentiality and pre-publicity policy may be found here <http://www.nature.com/authors/policies/confidentiality.html> Privacy Policy | Update Profile
DISCLAIMER: This e-mail is confidential and should not be used by anyone who is not the original intended recipient. If you have received this e-mail in error please inform the sender and delete it from your mailbox or any other storage mechanism. Springer Nature Limited does not accept liability for any statements made which are clearly the sender's own and not expressly made on behalf of Springer Nature Ltd or one of their agents. Please note that Springer Nature Limited and their agents and affiliates do not accept any responsibility

for viruses or malware that may be contained in this e-mail or its attachments and it is your responsibility to scan the e-mail and attachments (if any).

Reviewers' Comments:

Reviewer #1:

Remarks to the Author:

The authors have made a dedicated experimental effort to answer the question I raised. Based on their new data, it would seem that the lowered food intake cannot in itself explain their data. This was my major point and they have now included the new data in their paper as a supplemental figure.

My only comment is whether the authors - given their new data - should consider whether they mean (in the abstract middle) "...WAT browning was unlikely to have been caused by the reduced food intake, but was largely...." instead of the present formulation.

Reviewer #4:

Remarks to the Author:

I don't have further concerns.

Reviewer #5:

Remarks to the Author:

The authors have been highly, and altogether the manuscript reflects an incredible amount of work. I have no further comments.

REVIEWERS' COMMENTS:

Reviewer #1 (Remarks to the Author):

The authors have made a dedicated experimental effort to answer the question I raised. Based on their new data, it would seem that the lowered food intake cannot in itself explain their data. This was my major point and they have now included the new data in their paper as a supplemental figure. My only comment is whether the authors - given their new data - should consider whether they mean (in the abstract middle) "...WAT browning was unlikely to have been caused by the reduced food intake, but was largely..." instead of the present formulation.

Our response : Thanks for the reviewer's advice and we have made the modification in the abstract.

Reviewer #4 (Remarks to the Author): I don't have further concerns.

Reviewer #5 (Remarks to the Author): The authors have been highly, and altogether the manuscript reflects an incredible amount of work. I have no further comments.